# ABSINT-AI: Agentic Heap Abstractions for Abstract Interpretation

**Michael Wang**[1]  **Kexin Pei**[2]  **Armando Solar-Lezama**[1]

## Abstract

Static analysis typically relies on uniform, hard-coded heap abstractions. This limits their precision, especially in dynamic languages like JavaScript where heap structure is heterogeneous and difficult to analyze statically. We present ABSINT-AI |, a language-model-guided abstract interpreter that selects adaptive, per-object heap abstractions using high-level cues such as naming conventions and access patterns, avoiding brittle hand-engineered heuristics. The language model (LM) operates through a bounded interface and never directly manipulates program state, which preserves the soundness guarantees of abstract interpretation. We evaluate on a soundness-critical task: determining whether property accesses may dereference null or undefined references, a common requirement for enabling compiler optimizations such as check elimination. Evaluated on a benchmark of 53 JavaScript programs, ABSINT-AI reduces false positives by 13% over TAJS and over 50% relative to WALA, two state-of-the-art static analyzers, while preserving these guarantees.

## 1. Introduction

As dynamic languages like JavaScript find their way into more backend applications with strong performance requirements, there has been a growing interest in compiling them down to more optimal forms (ang; Serrano, 2022; Chandra et al., 2016). An important obstacle for these approaches is the difficulty of performing sound static program analysis on these languages due to their dynamic behavior and extensive use of complex heap allocated data (Feldthaus et al., 2013; Antal et al., 2023; Sridharan et al., 2012). This

[1]Massachusetts Institute of Technology, Cambridge, MA, USA [2]University of Chicago, Chicago, IL, USA. Correspondence to: Michael Wang <mi27950@mit.edu>, Kexin Pei <kpei@uchicago.edu>, Armando Solar-Lezama <asolar@csail.mit.edu>.

*Proceedings of the 43rd International Conference on Machine Learning*, Seoul, South Korea. PMLR 306, 2026. Copyright 2026 by the author(s).

is a problem because sound analysis is an essential element of compiler optimization (Hind, 2001; Schneck, 1973). Soundness ensures that the analysis captures all possible runtime behaviors of the program; without it, compilers cannot guarantee the safety of specific transformations.

A key challenge in sound and scalable static analysis for JavaScript is reasoning about the heap. JavaScript's dynamic object model allows programs to construct and mutate objects with unpredictable shapes, runtime-dependent fields, and implicit behavior tied to values stored within fields. Consider a typical loop that allocates multiple heterogeneous objects: some are short-lived wrappers, others are stable configuration records, and others may exhibit role-dependent behaviors encoded in field values. Traditional static analyses typically rely on uniform abstraction strategies, and often result in excessive over-approximation and imprecision. Constructing precise yet scalable heap abstractions is a major challenge for JavaScript due to its lack of static types and its permissive object model, and it remains a major bottleneck for static analysis frameworks.

In this paper, we introduce ABSINT-AI, an agentic framework that assists static analysis by performing heap abstractions. Our approach preserves the strong guarantees provided by traditional static analysis techniques while addressing some of their major limitations. Static analysis techniques analyze programs by treating them as sets of logical statements with well-defined semantics (Cousot & Cousot, 1977). This type of analysis can provide guarantees of soundness, but these methods leave out a lot of information, such as variable names, comments, general programming design patterns, and background knowledge. LMs on the other hand, are able to take advantage of this information very well, but lack the robustness of traditional static analysis. For example, changing variable names has been shown to have a drastic impact on model performance (Zeng et al., 2022; Srikant et al., 2021). ABSINT-AI combines the best of both worlds by using LMs to provide background information to a static analyzer without losing soundness guarantees.

The key design choice in ABSINT-AI is that it preserves the formal soundness guarantee of symbolic program analysis by constraining the LM to only choose from a predetermined set of *sound abstraction strategies* and decide *where* to apply abstractions. As a result, ABSINT-AI bounds

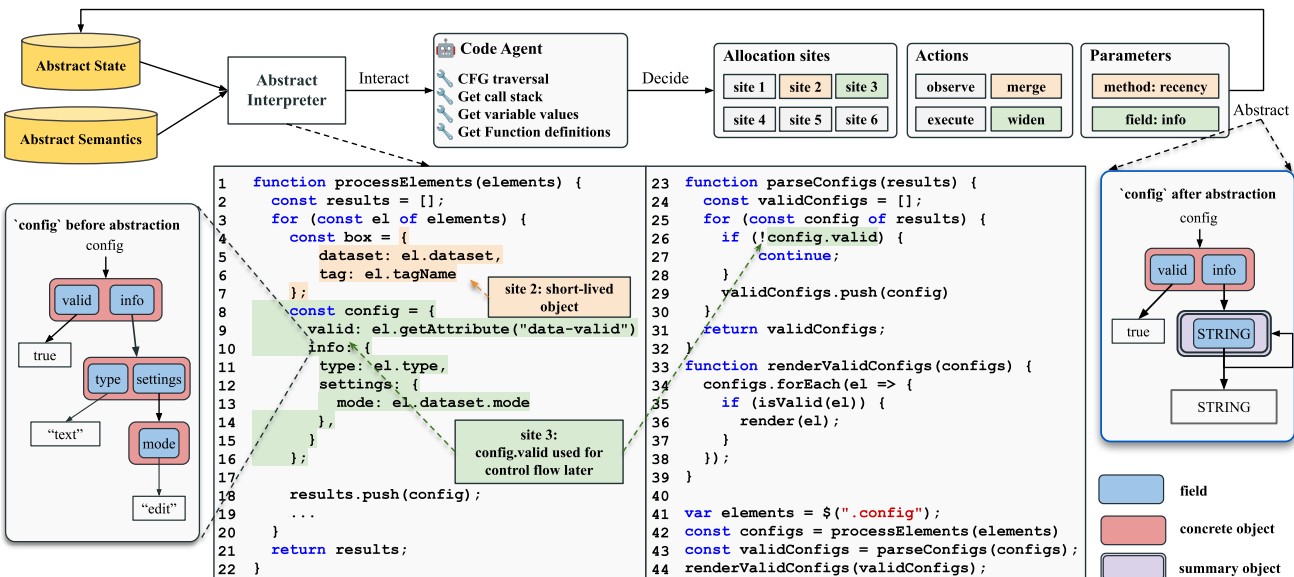

*Figure 1.* When ABSINT-AI encounters an unbounded loop, it suspends analysis and interacts with the language model agent for abstraction decisions. The agent selects a recency abstraction for `box` and a field-based widening for the `info` field of `config`, preserving relevant structure while ensuring convergence. A concrete instance of `config` is shown on the left, with its abstracted form on the right. These per-allocation-site abstraction decisions guide the analysis to a sound fixpoint.

the (inevitable) LM errors to only increased false positives (due to the aggressive abstraction decision) or slow down the convergence of the analysis (reduce to the precise but expensive analysis) without compromising the soundness.

Specifically, ABSINT-AI consists of a custom static analysis pipeline that invokes an agentic LM framework at key decision points - most notably before fixpoint computations in unbounded loops, where the choice of abstraction heavily influences convergence and precision. At each such point, the agent inspects the current analysis state, including the heap, code, and abstraction history. Based on this inspection, it selects appropriate abstraction strategies for each allocation site, such as merging objects using recency-abstraction, field sets, or value similarity. If the available information is insufficient to make a confident decision, the agent can request additional targeted analysis by executing the loop body for more iterations to refine its understanding. This interactive, goal-directed behavior enables adaptive, context-sensitive abstraction decisions and also allows the abstractions themselves to reflect higher-level semantic concepts. For example, if objects contain a `role` field, the agent can select a value-sensitive abstraction that merges all "teachers" into one object and all "students" into another, allowing domain-specific concepts to guide the abstractions themselves.

We evaluate our approach on the downstream task of detecting accesses to non-existent object fields, a common source of runtime errors in JavaScript. We compare our system against WALA (Santos & Dolby, 2022) and TAJS (Jensen et al., 2009), two state-of-the-art static analysis frameworks

that are representative of conventional heap abstraction strategies. Our evaluation on 53 JavaScript programs — including real-world npm libraries up to 1,885 lines of code — shows that ABSINT-AI reduces false positives by 13% over TAJS and over 50% relative to WALA while maintaining soundness. These results highlight the benefit of adaptive, semantically informed heap abstractions in improving the practical effectiveness of sound JavaScript analysis.

## 2. Motivating Example

Static analyses rely on heap abstractions (summaries of sets of objects), to reason about dynamic, heap-manipulating programs. The precision of these abstractions has a huge impact: too coarse and the analysis produces spurious warnings; too fine and it may never converge.

Modern JavaScript programs often construct diverse heap objects with different structural patterns and semantic roles, even within the same control-flow context. A one-size-fits-all heap abstraction applied uniformly across the entire program can lead to loss of precision or unnecessary state explosion. Consider the example in Figure 1, where each iteration of processElements allocates two distinct objects: a short-lived wrapper (`box`), and a structured configuration object (`config`). Each of these demands a different abstraction strategy. For instance, box can be aggressively summarized without affecting precision, while `config` exhibits a fixed field structure where only a single field, `valid`, must remain precise for correct downstream control flow. While it is theoretically possible to hand-engineer

heuristics that assign abstraction strategies based on object structure or access patterns, doing so at scale quickly becomes brittle, complex, and difficult to maintain. To the best of our knowledge, existing analyses do not adapt their heap abstractions per object, due to the complexity and brittleness of manually encoding such decisions.

However, many real-world objects contain semantic hints in field names or surrounding code that indicate how they should be abstracted. For example, the field `valid` suggests that the `config` object encodes access control logic, which is later reflected in a guard on `config.valid`. These high-level concepts such as "valid" configurations are difficult to capture using purely syntactic heuristics or static types, but are easily interpretable by language models. An agentic abstraction strategy can leverage such semantic cues to select more appropriate abstractions: preserving distinctions between roles, or widening only fields relevant to the analysis domain. This enables adaptive precision where it matters, and aggressive summarization where it doesn't—leading to more efficient and accurate analyses.

In ABSINT-AI, a language model acts as an agent that guides heap abstraction dynamically over the course of the analysis. Returning to the example in Figure 1, the agent might decide to apply recency abstraction to the short-lived `box` object and a field-set abstraction to the structured `config` object (preserving only `config.valid`). These decisions are not hardcoded: the agent queries the analysis environment for relevant context (such as variable values and function definitions), and may request additional loop iterations to test its abstraction choices. Crucially, all semantics and state transitions are handled by a traditional abstract interpreter, ensuring that soundness is preserved. The agent's role is purely to steer how the heap is abstracted, enabling more precise and efficient analysis by tailoring abstraction to the semantics of the program.

## 3. Methodology

ABSINT-AI is based on traditional abstract interpretation, but queries an LM to decide how to merge summary nodes at key points in the analysis. We provide an overview of static program analysis and abstract interpretation in Appendix A. The workflow of ABSINT-AI can be found in Figure 1.

### 3.1. Abstract Interpretation

Abstract interpretation requires an abstract domain as well as modeling of the heap. In this section, we briefly describe our abstract domain, our two-level representation of the heap, and when we invoke the LM for summarization. The full analysis supports prototypal inheritance, recursion, loops, and closures. Additional details can be found in the appendix.

```
1   var global = 0;
2   var global_obj = {};
3   function inc_global() {
4       let obj = {f: 1};
5       obj.f += 1;
6       global = global + obj.f;
7   }
8   function access_obj() {
9       if (global > 10) {
10          var f = global_obj.foo.bar; // bug
11      }
12  }
13  var btn1 = document.createElement("button");
14  var btn2 = document.createElement("button");
15  btn1.addEventListener("click", inc_global);
16  btn2.addEventListener("click", access_obj);
```

*Figure 2.* `inc_global` needs to be run at least 10 times before the bug on line 11 is triggered.

**Abstract Domain.** Our abstract domain keeps track of heap objects using concrete nodes and summary nodes. Summary nodes represent a set of possible concrete nodes.

Each node is a dictionary from primitive or abstract values to other values. Our domain of primitive values is based off of TAJS (Jensen et al., 2009), one of the first abstract interpretation based analyses for Javascript. The abstract domain and transfer functions are fixed; the LM agent does not alter the semantics of the analysis. Its role is limited to guiding when and where widening and merging operators are applied. Additional details on our abstract domain can be found in the appendix. The most important runtime decision of ABSINT-AI is deciding when summarize heap nodes. We keep two separate heap structures, referred to as the local heap and global heap.

**Local heap.** The local heap is used for precise representation for objects within local procedures, such as a local object allocation in a function call. It is flow-sensitive (Kildall, 1973), taking into account the order of statements. For example, in Figure 2, `obj` on line 4 is tracked in the local heap.

**Global heap.** The global heap is a much less precise representation for objects that are accessed and manipulated by multiple functions. The global heap captures all possible relationships between globally visible objects at any point in the execution. The global heap is motivated by flow-insensitive analysis (Weihl, 1980; Cousot & Cousot, 1977). This has two benefits: (1) It is much cheaper, as we don't have to keep track of a separate heap for each program location, and (2) it allows different functions to be analyzed independently; the global heap considers all the possible heap states at the point when the function is invoked, and the analysis of the function can reveal if any additional relationships need to be added to the global heap. Summarization only happens in the global heap.

We draw a distinction between the local and global heap because JavaScript programs tend to be reactive, with

execution driven largely by external events. This has important implications for analysis, as the analysis can't assume the program will simply execute starting at the beginning from a well defined initial state. Take the example in Figure 2, where `inc_global` is invoked by an event handler and must be executed at least 10 times in order to trigger the bug on line 11. Keeping two separate heaps allows us to to track global dependencies while not losing precision for local procedures.

**Agent Invocation.** A key challenge in abstract interpretation is to reach a fixpoint without losing too much precision when analyzing potentially unbounded loops. Because fixpoint computation requires merging abstract states across iterations, the choice of how to abstract heap objects allocated within the loop has a direct impact on both the precision and termination of the analysis.

Take the example in Figure 1. There are two objects, `box` and `config`. Each loop iteration allocates two objects: `box`, which is short-lived and well-suited to recency abstraction, and `config`, which contains a critical field (`valid`) that must remain precise. A uniform abstraction by allocation site would collapse these distinctions, introducing spurious behaviors. ABSINT-AI addresses this by invoking the LM agent at unbounded loops to choose abstraction strategies per object, balancing semantic precision with soundness and convergence. The agent is only invoked at unbounded loop joins, not at if–then–else merge points. Conditional branches use standard abstract joins and do not require agent intervention.

## 3.2. Agentic Heap Abstractions

The agent in our framework serves as an interactive component embedded within the analysis loop. Its role is to select heap abstraction strategies, but unlike a static classifier, it behaves as an agent that operates under partial information and interacts with its environment to gather context before acting.

The agent is not invoked as a one-shot oracle. Instead, it operates as a environment-interacting agent that gathers information over time. To make informed abstraction decisions, the agent interacts with the abstract interpreter and the abstract state to selectively gather semantic information from the program. Rather than exposing the entire program or heap state, which would overwhelm the agent and obscure the relevant context, we treat the interpreter as a queryable environment. This avoids a common challenge in machine learning for code: programs often contain far more information than an LLM can meaningfully process, especially in settings with deep heap structure.

The agent's outputs are limited to a predefined set of sound abstraction strategies, and it never directly manipulates

---

**Algorithm 1** Agentic Heap Abstraction Algorithm

---

**Require:** Loop $\mathcal{L}$, Analysis state $\mathcal{S}$, Allocation Sites $\mathcal{A}$
1: $b \leftarrow 0$ {Interaction counter (queries + executions)}
2: $\mathcal{A}' = \text{NONE}$
3: **while** $b < \text{budget}$ **do**
4:     Agent selects action $a \in \{\text{INFO}, \text{EXEC}, \text{SELECT}\}$
5:     **if** $a = \text{INFO}$ **then**
6:         Agent queries $\mathcal{S}$ for program information
7:         $b \leftarrow b + 1$
8:     **else if** $a = \text{EXEC}$ **then**
9:         Abstract Interpreter executes one iteration of the loop
10:         $b \leftarrow b + 1$
11:         continue
12:     **else if** $a = \text{SELECT}$ **then**
13:         Agent selects sites $\mathcal{A}' \subseteq \mathcal{A}$ to abstract
14:         break
15:     **end if**
16: **end while**
17: **if** $\mathcal{A}' = \text{NONE}$ **then**
18:     Agent selects $\mathcal{A}' \subseteq \mathcal{A}$ to abstract
19: **end if**
20: **for** $a_i \in \mathcal{A}'$ **do**
21:     Agent selects (Strategy, Parameters)
22:     Updated mapping in $\mathcal{S}$ from $a_i$ to strategy for $\mathcal{L}$
23: **end for**

---

program state or executes code. The underlying abstract interpreter remains responsible for all semantic computation and fixpoint reasoning. This architectural separation allows us to embed an adaptive, learning-driven agent within a sound static analysis framework—enabling high-level decision-making informed by context and semantics, while preserving formal correctness guarantees.

**Agent Interaction**. The agent is initialized with the current abstract state, including visible variables, relevant allocation site data, and any previously encountered heap shapes. It then enters an interactive decision-making loop. During this loop, the agent can issue queries to the abstract state for more information, such as requesting variable values, inspecting function definitions, or examining the heap shape. If the available information is insufficient, the agent may also postpone its decision making by requesting additional abstract loop iterations, allowing it to observe how the heap evolves over time. This enables the agent to defer commitment while gathering contextual evidence. We experimented with providing the full program and abstract state directly in the prompt, but the abstract heap often exceeded the model's context window for larger or deeply nested programs. To ensure stable, reproducible behavior,

the agent instead accesses information incrementally through `INFO` queries, retrieving only the specific variable or function summaries needed for each decision.

The interaction is bounded: the agent operates under a fixed query and iteration budget to ensure termination. Once satisfied, the agent returns a set of abstraction directives, specifying how the interpreter should merge and widen objects associated with each allocation site. The interpreter then executes the loop abstracting the heap as directed by the agent. If the abstract state does not reach a fixpoint within five iterations, it re-queries the agent for new abstraction strategies. Algorithm 1 contains a detailed description of our procedure.

The agent performs two decision stages:

**1. Selecting which allocation sites to summarize.**

- At each loop iteration, the interpreter identifies allocation sites whose abstract states changed.
- The agent receives a prompt containing:
    - The loop body and relevant code snippet.
    - A summary of changed allocation sites (object structures, points-to sets).
- The agent may issue up to 10 **read-only tool queries** to the analyzer (variable inspection, function lookup, or one extra abstract loop iteration) before selecting sites/strategies. Each query returns a result to the agent, which may then issue further queries or proceed to choose sites/strategies.
- Finally, the agent selects a set of allocation sites to summarize or merge, ensuring convergence before the next iteration.

**2. Choosing a merging strategy and widening strategy for each selected site.**

- For every selected allocation site, the interpreter asks the agent to choose one of the predefined parameterizable **merging strategies** for that site.
- The agent picks among them using natural-language cues from code and variable names.
- After picking a merging strategy, the interpreter asks the agent to choose one of the predefined parameterizable **widening strategies**.

Every action the agent can take is predefined, finite, and sound—it cannot invent new abstractions, only select among existing ones—and all interactions are deterministic within the interpreter.

**Read-only Tool Queries**. The agent can interact with ABSINT-AI via a small set of deterministic, read-only tool calls. Each call returns a structured result to the agent; the agent may use this result to decide whether to issue another query or to commit to an abstraction decision. We cap the number of tool calls per decision to keep the interaction bounded and reproducible. These queries are targeted: the agent specifies which variable/function to inspect, rather than receiving a full dump of the environment. The available tool calls are:

- **Variable inspection**: The agent requests the abstract value of a named in-scope variable (e.g., x).
- **Function introspection**: The agent requests the definition (or summary) of a named local function in scope (e.g., f).
- **Abstract loop step**: The agent requests exactly one additional abstract iteration and receives an updated summary of the changed allocation sites / relevant heap deltas.

These interactions allow the agent to incrementally reduce uncertainty and focus attention on semantically meaningful heap behaviors without drastically increasing the input size. In particular, loop execution supports deliberate abstraction delay, giving the agent a richer view of program dynamics before committing to a strategy.

**Merging Strategies.** Once the agent has identified which allocation sites require abstraction, it selects a merging strategy for each. This determines how objects allocated at that site are grouped during join operations. The agent chooses from the following predefined strategies:

- **Allocation-site merge**: Collapses all objects created at the same program location into a single abstract object.
- **Recency merge**: Preserves the most recently allocated object at that site; merges older instances.
- **Field-sensitive merge**: Groups objects with the same fields.
- **Role-based merge**: Partitions objects based on semantically meaningful field values (e.g., role), allowing distinctions like "student" vs. "teacher" to be preserved.

In particular, role-based merging requires semantic understanding of field names and value meanings; it is very difficult to implement role-based merging using purely symbolic techniques. Identifying that a specific field should guide abstraction boundaries is often a decision that depends on natural language cues and program intent.

After selecting a merging strategy for an allocation site, the agent also specifies a widening strategy. Widening determines how abstract heap objects are generalized over time as they are revisited across loop iterations. The agent chooses from the following strategies:

- **Field-set widening**: widen a selected subset of fields, leave the others concrete.
- **Field merging**: Merge the fields together, and select another widening strategy for the values. This is for handling infinitely growing objects.

```
1  let userId = 100; // abstracted to NUMBER.
2  let names = {100: "Jane"};
3  names[userId]; // False positive
```

*Figure 3.* False positive due to `userId` getting abstracted to the abstract NUMBER type.

- **Full widening**: recursively widen the entire object into a single shape.
- **Depth-based widening**: Collapse structures beyond a fixed depth threshold

These strategies allow the agent to control the granularity of abstraction per object: preserving precise structure where it matters while widening aggressively in parts of the heap that are less semantically relevant. As with merging, widening strategies are selected per allocation site and parameterized to balance precision with scalability.

### 3.3. Downstream Task

As a downstream task to test the precision of ABSINT-AI, we detect the following situations (1) accessing a property of `null` or `undefined` and (2) reading an absent property of an object.

Abstracting unnecessarily can lead to false positives. Take the example in Figure 3. If `userId` on line 1 gets abstracted to the abstract NUMBER type, then the object access on line 3 is reported as a possible read of an absent property. `userId` could take the value of all possible numbers, but `names` only has the the property `100`.

**Intersection of multiple runs.** Different abstraction choices in a program can lead to different sets of reported bugs. For example, when analyzing the program in Figure 3, ABSINT-AI may choose to abstract the `userId` field in some runs but leave it concrete in others. This variation can affect which false positives are reported. However, because each run is individually sound, any bug that does not appear in *any* run is guaranteed not to be real. This allows us to improve precision by taking the intersection of reported bugs across multiple runs (similar in spirit to self-consistency approaches (Wang et al., 2022b)) while preserving full soundness.

## 4. Evaluation

Our evaluation focuses on two key questions: (1) How does ABSINT-AI perform compared to existing static analysis tools? (2) What is the contribution of LM-guided abstraction selection relative to fixed symbolic strategies? To answer these, we compare against two established baselines (TAJS and WALA), conduct targeted ablations isolating the role of the LM, and present a case study demonstrating the system's ability to preserve meaningful heap structure.

### 4.1. Baselines

**TAJS**. TAJS (Type Analysis for JavaScript) performs flow-sensitive, context-sensitive, and partially path-sensitive static analysis of JavaScript programs (Jensen et al., 2009). TAJS is based on abstract interpretation, including specialized heap abstractions such as allocation-site abstraction and recency abstraction, to model JavaScript's dynamic object behavior.

**WALA**. WALA (T. J. Watson Libraries for Analysis) is a general-purpose static analysis framework that supports multiple languages, including JavaScript (Santos & Dolby, 2022). Unlike TAJS, WALA is not based on abstract interpretation and performs flow-insensitive heap analysis, using a combination of allocation-site abstraction and context-sensitive pointer analysis.

**Symbolic ABSINT-AI**. We also include a baseline that runs ABSINT-AI using a fixed abstraction configuration without LM guidance. This baseline selects a conservative widening strategy across all allocation sites, simulating how our analysis would perform without agentic control. It serves to isolate the contribution of the LM-driven adaptivity from the underlying analysis framework. Symbolic ABSINT-AI begins with recency-based merging and a depth-1 field-sensitive abstraction. If the loop fails to converge within 50 iterations, it switches to widening the entire object while maintaining recency-based merging. If convergence still fails after another 50 iterations, it falls back to a fully allocation-site-based abstraction.

**Dataset.** To evaluate our approach, we curated a benchmark of 53 self-contained JavaScript programs drawn from the Big Code dataset (Raychev et al., 2016), the V8 benchmark suite, GitHub, and 23 real-world open-source npm libraries, including PapaParse (CSV parser, 1,650 LOC), Showdown (Markdown converter, 1,680 LOC), Validator.js (string validation, 1,885 LOC), and JSBigNum (arbitrary-precision arithmetic, 1,357 LOC). In total, 9 benchmarks exceed 1,000 LOC and 6 exceed 1,500 LOC. We filtered programs that did not use builtins excessively, as this greatly increases the imprecision of the analysis (`Math.floor`, for example, requires modeling the `Math` library to analyze precisely). These require substantial modeling effort and introduce orthogonal complexity. We also excluded object-oriented programs that rely heavily on classes and `let` statements, since TAJS and WALA do not support JavaScript features after ES2015. A detailed description of the dataset can be found in the Appendix.

### 4.2. Performance

We evaluate ABSINT-AI using seven language models spanning multiple providers and capability tiers: GPT-4o-

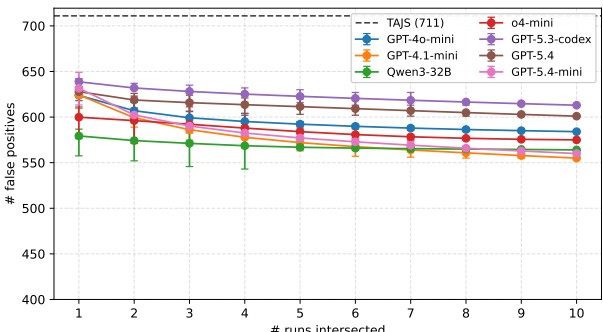

*Figure 4.* Running multiple times and taking the intersection of the reported bugs allows us to improve precision while maintaining soundness.

*Table 1.* Overall performance on the 53-benchmark suite (10 runs per model). #FP stands for False Positives; lower is better. TAJS crashes on 11/53 benchmarks; for those programs we conservatively assign it ABSINT-AI's own FP count.

|  | Model | # FP↓ | % Improve |
|---|---|---|---|
| Baselines | TAJS | 711 | 0% |
|  | WALA | 1327 | -86.6% |
|  | Symbolic ABSINT-AI | 820 | -15.3% |
| Mean | GPT-4o-mini | 628.5 | 11.6% |
|  | GPT-4.1-mini | 618.7 | 13.0% |
|  | Qwen3-32B | 616.9 | 13.2% |
|  | o4-mini | 631.8 | 11.1% |
|  | GPT-5.3-codex | 638.0 | 10.3% |
|  | GPT-5.4 | 638.1 | 10.3% |
|  | GPT-5.4-mini | 640.3 | 10.0% |
| Intersection | GPT-4o-mini | 584 | 17.9% |
|  | GPT-4.1-mini | 555 | 21.9% |
|  | Qwen3-32B | 564 | 20.7% |
|  | o4-mini | 573 | 19.4% |
|  | GPT-5.3-codex | 613 | 13.8% |
|  | GPT-5.4 | 601 | 15.5% |
|  | GPT-5.4-mini | 561 | 21.1% |
|  | Full Intersection | 499 | 29.8% |

mini, GPT-4.1-mini, Qwen3-32B, o4-mini, GPT-5.3-codex, GPT-5.4, and GPT-5.4-mini. To compare against TAJS and WALA, we measure the number of (1) possible accesses to a property of null or undefined or (2) possible reads of an absent property of an object. In this setting, lower values indicate greater precision, reflecting fewer spurious results caused by imprecise heap abstraction. We run ABSINT-AI 10 times across our benchmark per model and report the mean results in Table 1.

ABSINT-AI reports fewer false positives than either baseline, reducing total false positives by roughly 13% over TAJS and 53% over WALA on the strongest model. The Symbolic ABSINT-AI baseline (which uses our abstraction strategies but with a fixed widening schedule rather than LM-driven selection) produces 15% more false positives than TAJS, showing that the underlying abstraction framework alone does not match TAJS without LM guidance.

**Model comparison.** All seven models perform comparably,

with mean false positives spanning a narrow range from 616.9 (Qwen3-32B) to 640.3 (GPT-5.4-mini) — a spread of less than 4% of the lowest value. The older models (Qwen3-32B, GPT-4.1-mini, GPT-4o-mini, and o4-mini) perform marginally better than the GPT-5 family. Tool-use behavior varies widely and may be one contributing factor: GPT-4.1-mini, Qwen3-32B, and GPT-4o-mini invoke the abstract_loop_step tool frequently (333, 2,710, and 641 times respectively), while the remaining models invoke it sparingly (5 for o4-mini, 23 for GPT-5.4, and 0 for GPT-5.3-codex and GPT-5.4-mini). Our prompt discourages repeated loop execution to keep latency bounded, which may be interpreted more literally by some models than others. Tuning prompts to elicit more loop exploration is a natural avenue for future work.

**Intersection.** As described in Section 3.3, one benefit of maintaining soundness is that we can safely take the intersection of reported errors across multiple runs, improving precision without risking missed bugs. Figure 4 shows the effect of taking intersections across runs. The language model makes different abstraction decisions on different runs, leading to partially overlapping sets of reported warnings; intersecting them substantially reduces false positives. Across our 10 runs with GPT-4.1-mini, the intersection reduces the FP count from 618.7 (mean) to 555, an improvement of approximately 10% over a single run. We find that intersecting the top 3–4 runs gives the steepest improvement, with diminishing returns after 6 runs.

**Run time.** ABSINT-AI is slower than TAJS and WALA, primarily due to our prototype implementation in Python, whereas both TAJS and WALA are written in Java. Each run on the full benchmark suite takes approximately 500 seconds with GPT-4.1-mini, of which roughly 47% is LM interaction (network latency and inference time) and the rest is interpreter overhead. Because runs are independent, all 10 intersection runs can be launched in parallel, so the full intersection still completes in roughly 500 seconds of wall-clock time. In contrast, TAJS and WALA complete their analysis in approximately 20 seconds.

### 4.3. Ablations

**Ablation with symbolic abstractions.** To isolate the contribution of LM-guided abstraction selection, we compare ABSINT-AI to a purely symbolic variant that uses the same abstraction strategies without LM guidance, following the three-tier cascade described in Section 4 (recency merging with depth-1 widening, then recency merging with full-object widening, then allocation-site merging). Each tier is given 50 iterations to reach a fixpoint before escalating. If the analysis still does not converge after all three tiers within a 20-minute budget, we terminate and collect any warnings

```
1  var cell_state = [
2    [0, 1, 0],
3    [0, 1, 0],
4    [0, 1, 0]
5  ]
6  var n = parseInt($("#iterations"));
7  for (var i = 0; i < n; i++) {
8    cell_state = newGeneration(cell_state);
9  }
```

*Figure 5.* A snippet from Conway's Game of Life.

reported up to that point; 8 of 53 benchmarks required this fallback. As reported in Table 1, this symbolic variant produces 820 false positives — 15.3% more than TAJS. Despite using the same underlying abstraction strategies, the fixed cascade cannot match the LM's ability to choose per-allocation-site strategies based on program context.

**Ablation with non-agent LLM.** We also compare against a variant that uses the same language model in a non-agentic, single-shot setting. In this configuration, the model is prompted to select abstraction strategies directly, without the ability to query the interpreter, inspect intermediate state, or request additional loop iterations. Table 2 reports the results. The full agentic configuration reduces mean false positives from 645.7 to 618.7 (4.2%) and intersection false positives from 572 to 555 (3.0%).

*Table 2.* Non-agent ablation on the 53-benchmark suite (GPT-4.1-mini, 10 runs). The non-agent variant selects abstraction strategies in a single shot without environment queries.

| Variant | Mean FP | Intersection FP |
|---|---|---|
| ABSINT-AI (full agent) | 618.7 | 555 |
| No-agent (direct prediction) | 645.7 | 572 |

**Per-strategy ablation.** We removed each merging and widening strategy individually and re-ran ABSINT-AI to measure each strategy's contribution. Table 3 shows that every strategy contributes: removing `field_value` widening or `recency` merging causes the largest increases in false positives, while removing `depth` widening or `role` merging produces smaller — but still measurable — effects.

*Table 3.* Per-strategy ablation on the 53-benchmark suite (GPT-4.1-mini, 10 runs). Each row removes one strategy from the menu available to the LM.

| Removed Strategy | Mean FP | Δ vs Full |
|---|---|---|
| (none — full system) | 618.7 | — |
| No recency (merge) | 637.6 | +18.9 |
| No role (merge) | 630.0 | +11.3 |
| No `field_value` (widen) | 642.5 | +23.8 |
| No depth (widen) | 627.8 | +9.1 |

### 4.4. Case Study on Conway's Game of Life

To illustrate the benefits of agent-guided abstraction, we present a case study from our benchmark based on Con-

way's Game of Life in Figure 5. The `cell_state` variable represents a 3×3 grid of integers, updated over `n` iterations by the `newGeneration` function. While the contents change, the structure remains fixed across iterations; a property inherent to the game's rules. ABSINT-AI identifies that only the integer values need to be abstracted, preserving the shape of the array and producing a precise heap abstraction.

In contrast, symbolic baselines often over-abstract the structure itself, prematurely merging array shapes and losing row-level distinctions. This highlights how the agent draws on both program syntax and semantic cues such as common data patterns to guide more precise abstraction decisions.

## 5. Related Work

**LMs in program analysis.** LMs have been applied to a wide range of program analysis tasks, including type inference, fuzzing, vulnerability and resource leak detection, code summarization, and fault localisation (Peng et al., 2023; Wei et al., 2023; Wang et al., 2023b; Xia et al., 2024; Yang et al., 2023b;a; Deng et al., 2023; Mathews et al., 2024; Liu et al., 2023; Wang et al., 2023a; Mohajer et al., 2023; Cai et al., 2023; Geng et al., 2024; Ahmed et al., 2024; Wang et al., 2022a; Wu et al., 2023). However, none have been applied to static analysis while preserving soundness guarantees. More recently, several neurosymbolic approaches combine static analysis with LMs: LLift (Li et al., 2024a) filters false positives from UBITect (Zhai et al., 2020), IRIS (Li et al., 2024c) augments CodeQL (Avgustinov et al., 2016) for taint analysis, and InferROI (Wang et al., 2024) detects resource leaks in Java programs. While effective at improving precision, all of these systems sacrifice soundness once neural predictions are introduced.

**Program analysis for Javascript.** Much prior work on JavaScript analysis has focused on unsound but pragmatic tools for bug finding and security. These tools aim to detect likely vulnerabilities or errors in real-world programs, often trading soundness for scalability and precision (Li et al., 2022; Fass et al., 2019; Kang et al., 2023; Yu et al., 2023; Guo et al., 2024; Kang et al., 2025). While effective for finding particular security issues in practice, these approaches do not provide soundness guarantees. As a result, they are not suitable for many downstream tasks that depend on full program coverage, such as compiler optimizations or transformations, where missing even a single feasible behavior can invalidate correctness. Our work, by contrast, maintains the formal soundness of abstract interpretation while improving its precision via adaptive heap abstraction.

**LMs in sound reasoning.** Machine learning has been used to guide compiler optimization selection (Ansel et al., 2014; Huang et al., 2019), proof search and theorem proving (Bansal et al., 2019), as well as in program

synthesis (Li et al., 2024b) and SAT/SMT solving (Ganesh et al., 2022), where learned components suggest strategies or rule orderings without affecting overall soundness. In contrast, abstract-interpretation-based program analysis forms a distinct line of work, traditionally relying solely on manually designed heuristics for abstraction and widening. Singh et al. (Singh et al., 2018) use reinforcement learning to select abstract transformers in numerical domains (Polyhedra) for speed (up to 515x) while maintaining precision — similar in using learned policies to guide analysis, but with hand-crafted features on numerical domains rather than LLM-guided heap abstraction. Wang et al. (Wang et al., 2025) selectively widen only variables in value-flow cycles, reducing analysis time by 41%. Dewey et al. (Dewey et al., 2015) parallelize JavaScript abstract interpretation across allocation contexts to reduce wall-clock time, orthogonal to our precision-focused approach. To our knowledge, no prior system has incorporated large language models or other ML components into this framework while preserving soundness. Our method is the first to do so by constraining the LLM to select among a fixed, verified set of abstraction operators within a sound abstract domain.

## 6. Limitations and Conclusion

**Scalability**. A limitation of ABSINT-AI is that it does not scale to large JavaScript codebases (e.g., 2,000+ lines). This is a broader issue with JavaScript static analysis: neither TAJS nor WALA converged on such programs in our experiments. The challenge stems from the dynamic and object-heavy nature of real-world JavaScript. While our agent-guided approach adds adaptivity, our prototype and reliance on whole-program analysis similarly limit scalability. Addressing this is an important direction for future work.

**Abstraction operator coverage**. Our menu of merging and widening strategies draws from established techniques (recency abstraction, allocation-site collapse) together with our LM-friendly strategies (role-based merging, depth-bounded widening). It does not, however, cover all useful abstractions — in particular, relational domains and shape analyses that track invariants across multiple heap cells fall outside the current menu. Incorporating richer strategies and giving the agent the ability to compose them is a natural extension.

In this work, we propose a method to augment static analyzers with an agentic LM for heap abstractions. We present ABSINT-AI and an evaluation showing that augmenting static analysis with LMs can have an improvement on the precision without losing soundness guarantees.

## Impact Statement

This paper presents work whose goal is to advance the field of Machine Learning. There are many potential societal consequences of our work, none which we feel must be specifically highlighted here.

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

# A. Background

**Static program analysis**. Static program analysis aims to reason about all possible executions of a program. A key property is *soundness*, meaning the analysis never misses a real bug (no false negatives). The tradeoff is *precision*: overly coarse reasoning introduces spurious warnings (false positives).

To ensure scalability, analyses use abstractions that merge unbounded program behaviors (e.g., integers, heap objects) into finite summaries. For heap-manipulating languages like JavaScript, this typically means summarizing many concrete objects into a smaller set of abstract objects. The challenge is choosing what to merge: aggressive abstraction hurts precision, while conservative abstraction may prevent convergence. Prior work (Kanvar & Khedker, 2016) has developed many hand-written heuristics for heap abstractions. Our approach replaces such heuristics with LM-guided, adaptive abstractions.

**Soundness and precision.** Traditional static program analysis is often split between sound and unsound analyses. Soundness is the quality of static analyzers which guarantees that the analysis models an *over-approximation* of the target program's behavior, but may model behaviors that do not actually occur in any execution. The *precision* of the analysis is the extent to which the analysis avoids such spurious results. In short, a program analysis is *sound* if there are no false negatives. A program analysis is *precise* if there are not many false positives.

**Abstractions in static analysis.** Static analysis algorithms achieve scalability and soundness by using *abstractions* in their analysis. Programs often manipulate unbounded resources (e.g., integers, heap structures). Abstractions merge a potentially infinite set of objects into a single *summary* object to ensure convergence and for scalability. A key challenge is choosing *what* to abstract in the target program to ensure convergence while retaining as much important information as possible. There has been a rich body of literature on improving precision and scalability of heap abstractions (Kanvar & Khedker, 2016). In this work, we use an LM to decide what should be abstracted in the target program.

**Abstract Interpretation.** Abstract interpretation is a framework for analyzing programs by soundly approximating their behavior through the use of an *abstract state* that summarizes the set of possible states that a program can be in at different points in the execution (Cousot & Cousot, 1977). For simple programs manipulating scalar values, the abstract state is usually a simple mapping from variable names to abstract values representing sets of numbers. For example, an integer variable may be assigned the abstract value POSITIVE, representing all positive integers, to indicate the fact that its concrete value is guaranteed to be a positive value on any execution of the program. Abstract interpretation works by interpreting the program using rules that describe how each operation available in the language transforms the abstract state into new abstract states. For example, a rule may indicate that the addition of two POSITIVE numbers always results in a positive number. Soundness of the analysis is guaranteed by ensuring the soundness of each individual rule; for programs with loops, the analysis needs to be executed iteratively, and the theory of abstract interpretation ensures that once the abstract states converge to a fixpoint, this fixpoint will be a sound representation of the set of possible states that any execution of the program can reach.

For heap manipulating programs, the abstract state must include an abstraction of the heap which represents all the possible states of the heap a program might exhibit at a given point in time (Sagiv et al., 1998). There is an extensive literature on heap abstractions (Kanvar & Khedker, 2016), but all of them have a few elements in common. One important element is the use of *summarization* to represent multiple objects which may be living in the heap at a given point in the execution as a single *summary object*. Summarization allows the analysis to use a bounded representation for the potentially unbounded set of objects that can live on the heap on any arbitrary execution. Traditional abstract interpretation frameworks rely on complex heuristics to determine when and how to introduce summary nodes during program analysis to allow the analysis to maintain precision while quickly converging to a reasonably sized representation of the abstract heap. Our goal for this work is to replace those heuristics with an LM which can take advantage of its background knowledge of concepts used in the code as expressed through variable names, field names and comments.

# B. Abstract Interpretation Details

## B.1. Analysis details

**Functions** In Javascript, functions are stored as objects on the heap. We include a `__code__` property storing the function body to be executed. At the beginning of the analysis, ABSINT-AI scans the entire program, and generates a *schema* for each function. The schema for each function contains which variables are local to the function and which variables are accessed by other functions. We refer to variables that are local as *private*, and variables that are accessed by other functions as *shared*.

Each time a function is executed, an environment is initialized according to the schema for that function. When a function is defined, is initialized with a `__hf__` field set to the current heap frame. The `__hf__` field is used to model scopes and closures. When the function returns, the stack frame $\sigma$ is popped from the stack, and the stack pointer is decremented.

**Scopes and Closures** Whenever a function is called, a new stack frame $\sigma$ is pushed, along with a corresponding heap frame. The stack pointer for the current stack frame is updated to point to $\sigma$. The private variables for that function are stored in the stack frame $\sigma$, and any shared variables are stored in the heap frame. The heap frame is initialized with a parent field `__parent__` which is used to model the scope chain. The `__parent__` field points to the `__hf__` field for the function being initialized.

To lookup a variable name in the environment, ABSINT-AI first checks the current stack frame. If it finds a value for the variable, it returns the value. If it doesn't, it checks the corresponding heap frame for the stack frame, and then follows the chain of `__parent__` pointers until it finds the variable.

**Recursion** ABSINT-AI keeps track of all functions that have been called but have not finished executing yet. Whenever it encounters a recursive call, ABSINT-AI sets the return value to a recursive placeholder and stores a hash of the function that is called. When the function returns, ABSINT-AI checks the return values and any allocated heap objects for recursive placeholders for the function and fills them in with the return values.

### B.2. Environment

In this section we describe how ABSINT-AI represents the abstract state. We define concrete and abstract values. $H_L$ refers to the concrete heap, $H_G$ refers to the global heap, and $\sigma$ refers to the stack. $\tau$ is an abstract type, $C$ refers to constants, $obj$ and $\widetilde{obj}$ refer to concrete and abstract objects. $val$ and $\widetilde{val}$ refer to the values that a variable can take.

$$
\begin{array}{rcl}
val & ::= & a \,|\, obj \,|\, \widetilde{val} \\
\widetilde{val} & ::= & C \,|\, \widetilde{a} \,|\, \tau \,|\, \widetilde{obj} \\
\tau & ::= & Bool \,|\, Null \,|\, Num \,|\, String \\
obj & ::= & \tau \rightarrow val \,|\, C \rightarrow val \\
\widetilde{obj} & ::= & \tau \rightarrow \widetilde{val} \,|\, C \rightarrow \widetilde{val} \\
H_L & ::= & a \rightarrow val \\
H_G & ::= & \widetilde{a} \rightarrow \widetilde{val} \\
\sigma & ::= & C \rightarrow val
\end{array}
$$

### B.3. Syntax

$$
\begin{array}{rcl}
op & ::= & + \,|\, - \,|\, \div \,|\, \cdot \,|\, ... \\
E & ::= & id \,|\, E.field \,|\, E[E] \,|\, foo(E) \,|\, E_1[E_2](E_3, E4, ...) \,|\, \text{function}(x_0, x_1, ...)\{S\} \\
& & |\, \text{new } foo(E_1, E_2, ...) \,|\, C \,|\, \{f : E\} \\
varDef & ::= & \text{var } id = E \,|\, \text{let } id = E \,|\, \text{const } id = E \\
Stmt & ::= & varDef \,|\, id = E \,| \\
& & E.f = E \,|\, E[E] = E \,|\, \text{def } foo(x_1, x_2, ..., x_n)\{Stmt\} \,| \\
& & \text{if } (E)\{Stmt\} \text{ else } \{Stmt\} \,|\, \text{class } foo\{Stmt\} \,| \\
& & \text{return } E \,|\, \text{for } (varDef; E; Stmt)\{Stmt\} \\
& & \text{for } (varDef \text{ in } E)\{Stmt\} \,|\, \text{while } (E)\{Stmt\} \,|\, Stmt; Stmt
\end{array}
$$

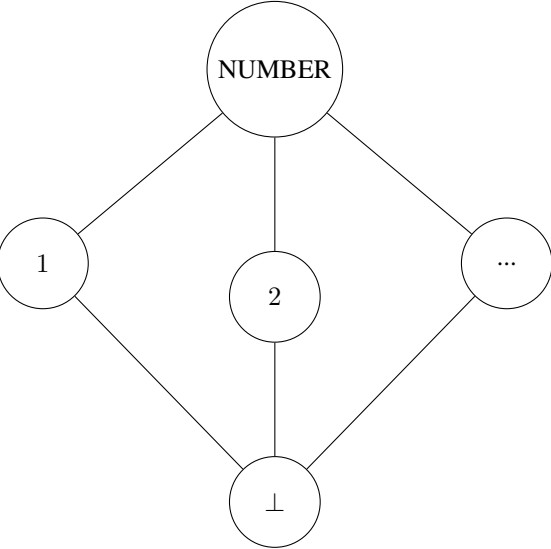

*Figure 6.* Number Lattice.

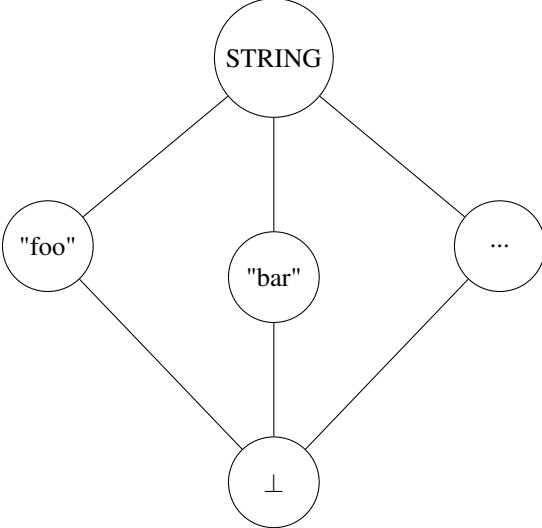

*Figure 7.* String Lattice.

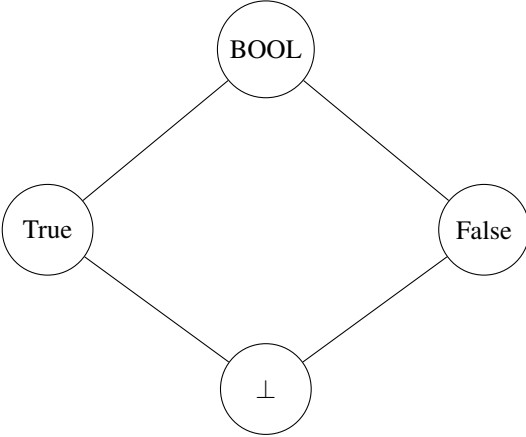

*Figure 8.* Boolean Lattice.

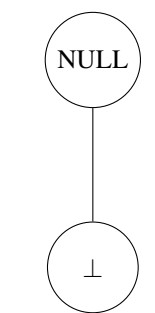

*Figure 9.* Null Singleton.

## B.4. Semantics

### B.4.1. FUNCTIONS

This section is several functions we use, such looking up a variable name and initializing a new schema for a function.

$$\text{lookup(id)} \qquad \frac{s \equiv \emptyset \quad \theta = \emptyset}{\langle lookup(H_L, H_G, s, id) \to \theta \rangle}$$

$$\frac{s \in H_L \quad id \in H_L(s) \quad \theta = s}{\langle lookup(H_L, H_G, s, id) \to \theta \rangle}$$

$$\frac{s \in H_G \quad id \in H_G(s) \quad \theta = s}{\langle lookup(H_L, H_G, s, id) \to \theta \rangle}$$

$$\frac{s \in H_L \quad id \notin H_L(s) \quad \theta = lookup(H_L, H_G, H_L(s).par, id)}{\langle lookup(H_L, H_G, s, id) \to \theta \rangle}$$

$$\frac{s \in H_G \quad id \notin H_G(s) \quad \theta = lookup(H_L, H_G, H_G(s).par, id)}{\langle lookup(H_L, H_G, s, id) \to \theta \rangle}$$

$$\text{initialize(schema)} \qquad \frac{H_L[a \mapsto \{schema.public, par \mapsto \sigma.hf\}] \quad \sigma'.\_secret \mapsto \{schema.secret\} \quad \sigma'.hf \mapsto a}{initialize(schema) \to H_L, H_G, \sigma :: \sigma'}$$

$$\text{return\_from\_schema} \qquad \frac{\sigma \equiv \sigma' :: v}{return\_from\_schema \to H_L, H_G, \sigma'}$$

### B.4.2. SMALL-STEP SEMANTICS

$$\langle H_L, H_G, \sigma, S \rangle \to \langle H'_L, H'_G, \sigma', S' \rangle$$

id
$$\frac{}{\langle H_L,H_G,\sigma,id\rangle \rightarrow \langle H_L,H_G,\sigma,lookup(id)\rangle}$$

E.field
$$\frac{\langle H_L,H_G,\sigma,E\rangle \rightarrow \langle H'_L,H'_G,\sigma',V\rangle}{\langle H_L,H_G,\sigma,\text{E.field}\rangle \rightarrow \langle H'_L,H'_G,\sigma',\text{get(V,field)}\rangle}$$

$E_1[E_2]$
$$\frac{\langle H_L,H_G,\sigma,E_2\rangle \rightarrow \langle H'_L,H'_G,\sigma',V_2\rangle \quad \langle H'_L,H'_G,\sigma',E_1\rangle \rightarrow \langle H''_L,H''_G,\sigma'',V_1\rangle}{\langle H_L,H_G,\sigma,E_1[E_2]\rangle \rightarrow \langle H'_L,H'_G,\sigma',\text{get}(V_1,V_2)\rangle}$$

$foo(E_0,E_1,...)$
$$\frac{\langle lookup(foo) \rightarrow V, V.\_\_type \equiv Function\rangle \quad \langle H_L,H_G,\sigma,E_0,E_1,...\rangle \rightarrow \langle H'_L,H'_G,\sigma',V_0,V_1,...\rangle}{\langle H_L,H_G,\sigma,foo(E_0,E_1,...)\rangle \rightarrow \langle H'_L[x_0 \mapsto V_0,x_1 \mapsto V_1,...],H'_G,\sigma',initialize(V.\_\_code);V.\_\_code\rangle}$$

$E_1[E_2](E_3,E_4,...)$
$$\frac{\langle H_L,H_G,\sigma,E_0,E_1,...\rangle \rightarrow \langle H'_L,H'_G,\sigma',V_0,V_1,V_2,...\rangle \quad \langle get(V_0,V_1) \rightarrow V, V.\_\_type \equiv Function\rangle}{\langle H_L,H_G,\sigma,foo(E_0,E_1,...)\rangle \rightarrow \langle H'_L[x_0 \mapsto V_0,x_1 \mapsto V_1,...],H'_G,\sigma'[this \mapsto V_0],V.\_\_code\rangle}$$

$function(x_0,x_1,...)\{S\}$
$$\frac{}{\langle H_L,H_G,\sigma,\text{function}(x_0,x_1,...)\rangle \rightarrow \langle H'_L[a' \mapsto \{...,prototype:a\},a \mapsto],H'_G,\sigma',a'\rangle}$$

new $foo(E_0,E_1,...)$
$$\frac{\langle lookup(foo) \rightarrow V\rangle \quad \langle V.\_\_type \equiv Class\rangle \quad \langle E_0,E_1,...\rangle \rightarrow \langle V_0,V_1,...\rangle}{\langle H_L,H_G,\sigma,\text{new}\,foo(E_0,E_1,...)\rangle \rightarrow \langle H'_L,H'_G,\sigma'[this \mapsto V],init();get(prototype(V),constructor)(V_0,V_1,...)\rangle}$$

$$\frac{\langle lookup(foo) \rightarrow V\rangle \quad \langle V.\_\_type \equiv Function\rangle \quad \langle E_0,E_1,...\rangle \rightarrow \langle V_0,V_1,...\rangle}{\langle H_L,H_G,\sigma,\text{new}\,foo(E_0,E_1,...)\rangle \rightarrow \langle H'_L,H'_G,\sigma',V.\_\_code(V_0,V_1,...)\rangle}$$

$\{f_1:E_1,f_2:E_2,...\}$
$$\frac{\langle H_L,H_G,\sigma,E_1,E_2,...\rangle \rightarrow \langle H'_L,H'_G,\sigma',V_1,V_2,...\rangle}{\langle H_L,H_G,\sigma,\{f_1:E_1,f_2:E_2,...\}\rangle \rightarrow \langle H_L[a \mapsto \{f_1:V_1,f_2:V_2,...,\_\_type:\text{object}\}],H_G,\sigma,a\rangle}$$

(var x = E)
$$\frac{\langle H_L,H_G,\sigma,E\rangle \rightarrow \langle H'_L,H'_G,\sigma',V\rangle \quad \theta = lookup(x) \quad \theta \in H_L \quad \text{fr} = H_L[\theta] \quad \text{fr'} = fr[id \mapsto V]}{\langle H_L,H_G,\sigma,x=E\rangle \rightarrow \langle H'_L[\theta \mapsto fr'],H'_G,\sigma',skip\rangle}$$

$$\frac{\langle H_L,H_G,\sigma,E\rangle \rightarrow \langle H'_L,H'_G,\sigma',V\rangle \quad \theta = lookup(x) \quad \theta \in H_G \quad \text{fr} = H_G[\theta] \quad \text{fr'} = fr[id \mapsto V \cup fr[id]]}{\langle H_L,H_G,\sigma,x=E\rangle \rightarrow \langle H'_L,H'_G[\theta \mapsto fr'],\sigma',skip\rangle}$$

(x.f = E)
$$\frac{lookup(x) \equiv a \quad \theta = H_L(a) \quad \langle H_L,H_G,\sigma,E\rangle \rightarrow \langle H'_L,H'_G,\sigma',V\rangle}{\langle H_L,H_G,\sigma,x.f=E\rangle \rightarrow \langle H'_L[\theta[f \mapsto V]],H'_G,\sigma',skip\rangle}$$

$$\frac{lookup(x) \equiv \widetilde{a} \quad \widetilde{\theta} = H_G(\widetilde{a}) \quad \langle H_L,H_G,\sigma,E\rangle \rightarrow \langle H'_L,H'_G,\sigma',V\rangle}{\langle H_L,H_G,\sigma,x=E\rangle \rightarrow \langle H'_L,H'_G[\widetilde{\theta}[f \mapsto V],\sigma',skip\rangle}$$

(x[E] = E')
$$\frac{lookup(x) \equiv a \quad \theta = H_L(a) \quad \langle H_L, H_G, \sigma, E, E' \rangle \to \langle H'_L, H'_G, \sigma', V, V' \rangle}{\langle H_L, H_G, \sigma, x[f] = E \rangle \to \langle H'_L[\theta[V \mapsto V']], H'_G, \sigma'], skip \rangle}$$

$$\frac{lookup(x) \equiv \widetilde{a} \quad \widetilde{\theta} = H_G(\widetilde{a}) \quad \langle H_L, H_G, \sigma, E, E' \rangle \to \langle H'_L, H'_G, \sigma', V, V' \rangle}{\langle H_L, H_G, \sigma, x = E \rangle \to \langle H'_L, H'_G[\widetilde{\theta}[V \mapsto V'], \sigma', skip \rangle}$$

(def foo$(x_0, x_1, ..., x_n)$\{Stmt\})
$$\frac{\theta = lookup(foo) \quad \theta \in \sigma}{\langle H_L, H_G, \sigma, x[f] = E \rangle \to \langle H_L[a \mapsto ..., prototype : a', a' \mapsto \{\}], H_G, \sigma[\theta \mapsto a], skip \rangle}$$

$$\frac{\theta = lookup(foo) \quad \theta \in H_L}{\langle H_L, H_G, \sigma, x[f] = E \rangle \to \langle H_L[a \mapsto ..., prototype : a', a' \mapsto \{\}, \theta \mapsto a], H_G, \sigma, skip \rangle}$$

$$\frac{\theta = lookup(foo) \quad \theta \in H_G}{\langle H_L, H_G, \sigma, x[f] = E \rangle \to \langle H_L, H_G[a \mapsto ..., prototype : a', a' \mapsto \{\}, \theta \mapsto \theta \cup a], \sigma, skip \rangle}$$

(x[E] = E')
$$\frac{lookup(x) \equiv a \quad \theta = H_L(a) \quad \langle H_L, H_G, \sigma, E, E' \rangle \to \langle H'_L, H'_G, \sigma', V, V' \rangle}{\langle H_L, H_G, \sigma, x[f] = E \rangle \to \langle H'_L[\theta[V \mapsto V']], H'_G, \sigma'], skip \rangle}$$

$$\frac{lookup(x) \equiv \widetilde{a} \quad \widetilde{\theta} = H_G(\widetilde{a}) \quad \langle H_L, H_G, \sigma, E, E' \rangle \to \langle H'_L, H'_G, \sigma', V, V' \rangle}{\langle H_L, H_G, \sigma, x = E \rangle \to \langle H'_L, H'_G[\widetilde{\theta}[V \mapsto V'], \sigma', skip \rangle}$$

if (E) \{ Stmt \}) else \{ Stmt' \}
$$\frac{\langle H_L, H_G, \sigma, E \rangle \to \langle H'_L, H'_G, \sigma', False \vee \emptyset \rangle}{\langle H_L, H_G, \sigma, \text{if}(E)\{Stmt\} \text{ else } \{Stmt;\} \rangle \to \langle H'_L, H'_G, \sigma' \rangle, Stmt \rangle}$$

$$\frac{\langle H_L, H_G, \sigma, E \rangle \not\to \langle H'_L, H'_G, \sigma', False \vee \emptyset \rangle}{\langle H_L, H_G, \sigma, \text{if}(E)\{Stmt\} \text{ else } \{Stmt'\} \rangle \to \langle H'_L, H'_G, \sigma' \rangle, Stmt' \rangle}$$

class foo$[M_1, M_2, ..., M_N]$
$$\frac{class\_obj = \{M_1, M_2, ..., M_N\}}{\langle H_L, H_G, \sigma, \text{class foo}[M_1, M_2, ..., M_N] \rangle \to \langle H_L[a \mapsto class\_obj], H_G, \sigma, skip \rangle}$$

(return E)
$$\frac{\langle H_L, H_G, \sigma, E \rangle \to \langle H'_L, H'_G, \sigma', V \rangle}{\langle H_L, H_G, \sigma, return E \rangle \to \langle H'_L, H'_G, \sigma'[returns \mapsto \sigma'[returns] \cup V], skip \rangle}$$

for ([let | var] id in E) \{ Stmt \}
$$\frac{\langle H_L, H_G, \sigma, E \rangle \to \langle H'_L, H'_G, \sigma', V \rangle \quad V.\_proto\_ \equiv \emptyset \quad isEmpty(V) \equiv True}{\langle H_L, H_G, \sigma, \text{for ([let | var] id in E) } \{Stmt\} \rangle \to \langle H'_L, H'_G, \sigma', skip \rangle}$$

$$\frac{\langle H_L, H_G, \sigma, E \rangle \to \langle H'_L, H'_G, \sigma', V \rangle \quad V.\_proto\_ \not\equiv \emptyset \quad isEmpty(V) \equiv True}{\langle H_L, H_G, \sigma, \text{for ([let | var] id in E) } \{Stmt\} \rangle \to \langle H'_L, H'_G, \sigma', \text{for ([let | var] id in V.\_proto\_) } \{Stmt\} \rangle}$$

$$\frac{\langle H_L, H_G, \sigma, E \rangle \to \langle H'_L, H'_G, \sigma', V \rangle \quad V \equiv X :: V' \quad varDef.type \equiv let}{\langle H_L, H_G, \sigma, \text{for (let id in E) } \{Stmt\} \rangle \to \langle H''_L, H''_G, \sigma'', \text{initialize(Stmt);let id=X;Stmt; for (let id in V') \{ Stmt \}} \rangle}$$

$$\frac{\langle H_L, H_G, \sigma, E \rangle \to \langle H'_L, H'_G, \sigma', V \rangle \quad V \equiv X :: V' \quad varDef.type \equiv var}{\langle H_L, H_G, \sigma, \text{for (let id in E) } \{Stmt\} \rangle \to \langle H'_L, H'_G, \sigma', \text{var id=X;Stmt; for (let id in V') \{ Stmt \}} \rangle}$$

while (E) $\{Stmt\}$
$$\frac{\langle H_L,H_G,\sigma,E\rangle \rightarrow \langle H_L',H_G',\sigma',V\rangle \quad V \in \text{Falsey}}{\langle H_L,H_G,\sigma,\text{while (E) }\{Stmt\}\rangle \rightarrow \langle H_L',H_G',\sigma',skip\rangle}$$

$$\frac{\langle H_L,H_G,\sigma,E\rangle \rightarrow \langle H_L',H_G',\sigma',V\rangle \quad V \notin \text{Falsey} \quad \langle H_L',H_G',\sigma',\text{Stmt;summarize()}\rangle \rightarrow \langle H_L',H_G',\sigma'\rangle}{\langle H_L,H_G,\sigma,\text{while (E) }\{Stmt\}\rangle \rightarrow \langle H_L',H_G',\sigma',skip\rangle}$$

$$\frac{\langle H_L,H_G,\sigma,E\rangle \rightarrow \langle H_L',H_G',\sigma',V\rangle \quad V \notin \text{Falsey} \quad \langle H_L',H_G',\sigma',\text{Stmt;summarize()}\rangle \rightarrow \langle H_L'',H_G'',\sigma''\rangle}{\langle H_L,H_G,\sigma,\text{while (E) }\{Stmt\}\rangle \rightarrow \langle H_L'',H_G'',\sigma'',\text{while (E) }\{Stmt\}\rangle}$$

## C. Implementation and Dataset

**Implementation.** We implemented ABSINT-AI in 8049 lines of Python, and use Espree (brettz9) to parse the Javascript into an AST. We conducted the experiments on a Linux server with two AMD EPYC 7763 64-Core Processors, 128 cores, 1024GB RAM, and 4 NVIDIA RTX 6000 Ada Generation GPUs.

## C.1. Dataset

*Table 4.* The 53 JavaScript programs in our benchmark suite.

| Program | #Lines | Description |
| --- | --- | --- |
| 2048.js | 234 | 2048 game implemented for the DOM. |
| binomial_heap.js | 199 | Binomial heap data structure. |
| bloom_filter.js | 135 | Bloom filter data structure. |
| bonzo.js | 395 | DOM utility library. |
| breakout_game.js | 210 | Breakout arcade game for the DOM. |
| breakout_game2.js | 159 | Alternative implementation of Breakout for the DOM. |
| bst.js | 78 | Binary search tree. |
| cbuffer.js | 163 | Circular buffer. |
| chatbot.js | 182 | Simple rule-based chatbot. |
| confetti.js | 393 | Confetti animations in the DOM. |
| ConwaysGameOfLife.js | 64 | Conway's Game of Life. |
| currencyFormatter.js | 1284 | Currency formatting library. |
| datepattern.js | 90 | Date string pattern matching. |
| denque.js | 175 | Double-ended queue. |
| event_emitter.js | 54 | Event emitter pattern. |
| graph.js | 160 | Graph data structure. |
| hash-map.js | 577 | HashMap implementation. |
| heap.js | 78 | Heap data structure. |
| html5-canvas-drawing-app.js | 440 | Drawing app in the DOM. |
| huffman.js | 114 | Huffman coding. |
| humanize_duration.js | 1614 | Humanize time durations as natural-language strings. |
| jsbn.js | 1357 | Arbitrary-precision arithmetic (JSBigNum). |
| jshashes.js | 1747 | Cryptographic hash functions. |
| json_parser.js | 165 | JSON parser. |
| levenshtein.js | 116 | Levenshtein edit distance. |
| linked_list.js | 159 | Linked list. |
| lru_cache.js | 128 | LRU cache. |
| matrix.js | 98 | Matrix operations. |
| mersenne.js | 65 | Mersenne Twister pseudorandom number generator. |
| minesweeper.js | 166 | Minesweeper game. |
| music_player.js | 196 | Music player in the DOM. |
| mustache.js | 618 | Mustache template engine. |
| navier-stokes.js | 385 | Simplified Navier–Stokes fluid simulation. |
| numeral.js | 1008 | Number formatting library. |
| papaparse.js | 1650 | CSV parser (PapaParse). |
| perlin.js | 111 | Perlin noise generation. |
| pong.js | 235 | Pong game in the DOM. |
| priority_queue.js | 117 | Priority queue. |
| pubsub.js | 66 | Publish–subscribe pattern. |
| quiz.js | 92 | Quiz / trivia logic. |
| router.js | 92 | URL router. |
| showdown.js | 1680 | Markdown-to-HTML converter (Showdown). |
| signature_pad.js | 453 | Canvas-based signature pad. |
| snake_game.js | 103 | Snake game in the DOM. |
| speakingurl.js | 1675 | URL slug generator. |
| splay.js | 406 | Splay tree implementation. |
| sudoku_solver.js | 109 | Sudoku solver. |
| tetris.js | 133 | Tetris game. |
| TimSort.js | 111 | Tim Sort algorithm. |
| trie.js | 120 | Trie data structure. |
| uaparser.js | 908 | User-agent string parser (UAParser). |
| validator.js | 1885 | String validation library (Validator.js). |
| vdom.js | 325 | Virtual DOM implementation. |

# D. Prompts

This section reproduces the prompts used by the LM agent. Tool schemas are documented inline with each prompt.

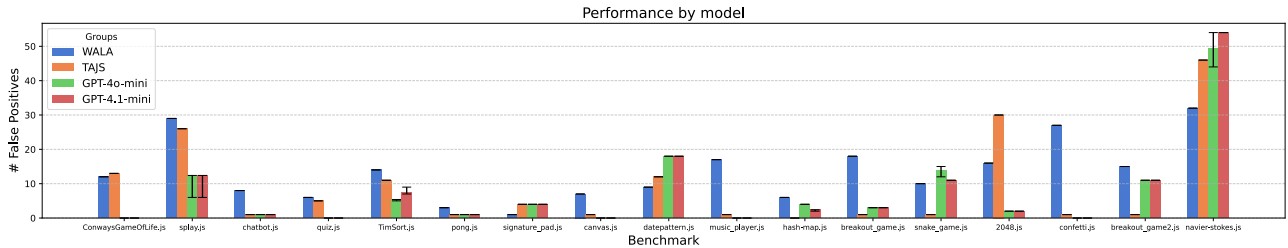

*Figure 10.* Performance per model on each benchmark program compared to WALA and TAJS.

**System prompt (agentic mode).** Instructs the agent to decide which heap allocation sites need summarization for loop convergence. The agent gathers information via tool queries before selecting sites, constrained to respond only via tool calls. A budget of 10 tool queries prevents excessive querying.

```
You are an expert static analysis assistant.

Your task is to decide which heap allocation sites should be summarized
    in order to ensure that the program's abstract interpretation converges at a loop.

If an allocation site produces
    objects whose values (e.g. fields or shapes) may grow or change across iterations,
    you should select that site for summarization. If an allocation site produces only one
    object or produces identical values each iteration, it may not need to be summarized.

You may only respond by calling
    a tool. Do not produce natural language explanations unless instructed to do so.

You must make decisions only
    after gathering the necessary information. Use the tools 'info_var', 'info_function
    ', and 'execute_loop' provided to inspect variables, look up function definitions
    , or execute the loop. You should use them to gather information before making a
    decision. If you decide to use these tools, do NOT summarize the allocation sites in
    the same iteration. you should only summarize allocation sites after you have gathered
    all the necessary information. Specifically, 'execute_loop' is ONLY used to gather
    information, nothing else. If you have executed the loop five times and you still don
    't have enough information, you should summarize the allocation sites anyway. Do NOT
    try to execute the loop more than five times in a row. If you do, you should summarize
    the allocation sites anyway. DO NOT REPEATLY EXECUTE THE LOOP. This is a waste
    of time and resources. You should only execute the loop once to gather information.
    If you need more information, you should use the 'info_var' or 'info_function' tools.

Be conservative
    in your choices --- avoid summarizing sites that do not contribute to divergence.
```

Tool schemas exposed in agentic mode:

```
select_allocation_sites(selected_sites: list[str])
    # Select allocation sites that should be summarized to ensure
    # convergence of the loop. selected_sites must be drawn from the
    # set of allocation sites changed in the current iteration.

info_var(var_name: str)
    # Get the current value or summary of a variable in the environment.

info_function(function_name: str)
    # Get the current definition of a function by name in the environment.

execute_loop()
    # Execute the loop body once to update the environment state.
```

**System prompt (no-agent ablation).** Same task description as the agentic prompt, but without access to environment

query tools — the agent must select allocation sites based solely on the source code context provided in the prompt.

```
You are an expert static analysis assistant.

Your task is to decide which heap allocation sites should be summarized
    in order to ensure that the program's abstract interpretation converges at a loop.

If an allocation site produces
    objects whose values (e.g. fields or shapes) may grow or change across iterations,
   you should select that site for summarization. If an allocation site produces only one
    object or produces identical values each iteration, it may not need to be summarized.

Be conservative
    in your choices --- avoid summarizing sites that do not contribute to divergence.
```

**Merging strategy prompt.** For each selected allocation site, the agent chooses among the merging strategies described in Section 3.3: `all`, `recency`, `role`, and `field_sensitive`. The prompt provides the surrounding code, loop body, and current abstract value. Placeholders in curly braces are substituted at runtime.

```
You are configuring the merge strategy for {allocation_site}. Here is the information,

=== Code ===
{code}

=== Loop Body ===
{loop_body}

=== Allocation Site Value ===
{allocation_site_value}

Choose a strategy for merging the values of this allocation site.

Valid options are: all, recency, role, field_sensitive.

- all: simply merge
    all possible addresses and primitives together into one mega-object. Do this if the
    object is not being re-allocated frequently, and is just one object being modified.
- recency: Keep the most recent values of the allocation site, and merge the old ones into
    a single summary node. Only do this is the object is being re-allocated frequently.
- role: merge
    objects that have a similar role. Specify the role as a field name, and all addresses
    that have the same value for that field will be merged together. For example, if you
    want to keep a separate abstract object for different user roles, or different types
    of AST nodes. If you choose this strategy, you MUST provide the field as a parameter.
- field_sensitive
    : merge objects that have the same set of field names. Objects with different fields
    stay separate (up to 8 groups). Good when objects at the same allocation site have
    structurally different shapes (e.g., a constructor that sometimes adds extra fields).
```

Tool schema:

```
select_merge_strategy(
    strategy: "all" | "recency" | "role" | "field_sensitive",
    field: str  # required only if strategy == "role"
)
```

**Widening strategy prompt.** For each selected allocation site, the agent chooses among the widening strategies: `none`, `depth`, `field_value`, and `all`. Same context as the merging prompt.

```
You are configuring the widening strategy for {allocation_site}. Here is the information,

=== Code ===
{code}
```

```
=== Loop Body ===
{loop_body}

=== Allocation Site Value ===
{allocation_site_value}

Choose a strategy for merging the values of this allocation site.

Valid options are: field_value, all, depth.

- field_value
    : widen the value for a few particular fields. You might do this if only a few
      fields are growing. Provide a space-separate list of field paths using dot notation.
- all: widen the entire thing. This is very imprecise, so use it sparingly.
- depth
    : widen all values after a particular depth. Provide the depth as an integer. If the
      depth is 1, it will widen all field values. If the depth is 2, it will find all field
      values 2 levels deep and widen them, etc. This is a good option if you have a lot
      of fields that are all changing. This is a very good one. REMEMBER DEPTH IS 1-BASED.
```

Tool schema:

```
select_widening_strategy(
    strategy: "field_value" | "all" | "none" | "depth",
    fields: list[str]  # dot-separated paths, used only if strategy == "field_value"
    depth: int         # used only if strategy == "depth", default 1
)
```

## E. Additional Experiments

### E.1. Query Budget Sensitivity

We varied the per-decision tool-query budget from 0 to 15 and re-ran ABSINT-AI on the full 53-benchmark suite (10 runs per setting). Table 5 shows that precision improves steadily as the budget grows, with diminishing returns past 10. A budget of zero corresponds to the non-agent variant: the LM must commit to abstraction strategies without inspecting the analysis state.

*Table 5.* Effect of the tool-query budget on mean false positives (53 benchmarks, 10 runs, GPT-4.1-mini).

| Budget | Mean FP |
|---|---|
| 0 | 645.7 |
| 1 | 628.5 |
| 3 | 633.0 |
| 5 | 631.7 |
| 7 | 621.4 |
| 10 (default) | 613.6 |
| 15 | 614.2 |

### E.2. Bug Injection

To verify that ABSINT-AI's soundness is preserved in practice (and that intersection across runs does not drop real bugs), we injected null-dereference and absent-property-access bugs into 4 benchmarks and ran ABSINT-AI 10 times on each. Every injected bug was detected in 100% of runs (42/42 total) and consequently appeared in every intersection. This is consistent with the soundness argument: each individual run over-approximates all reachable states, so any genuine bug must be reported in every run, and intersecting runs can therefore only remove false positives.

### E.3. Agent Memory on Re-query

In the submitted version, the agent does not receive context about its prior failed strategy choices when re-queried after a convergence failure. We implemented a variant that passes the agent its previous strategy selections and the convergence-

failure context on re-query. Table 6 shows the comparison over 10 runs on the 53-benchmark suite. The improvement is marginal, likely because LM sampling temperature already produces sufficiently diverse strategy choices across re-queries.

*Table 6.* Agent memory on re-query (53 benchmarks, 10 runs, GPT-4.1-mini).

| Variant | Mean FP | Intersection FP |
|---|---|---|
| Without memory | 618.7 | 555 |
| With memory | 614.6 | 557 |

### E.4. Behavior Statistics

To characterize how the agent uses the strategy menu in practice, we aggregated all 10 runs on the 53-benchmark suite for each model evaluated in Table 1. Tables 7, 8, and 9 report, respectively, the distribution of environment queries the agent issues before deciding, and the distributions of selected merging and widening strategies.

Query behavior varies dramatically across models (Table 7). Qwen3-32B leans almost entirely on abstract loop stepping (2,710 calls), while GPT-5.4 prefers variable inspection (2,084 calls). GPT-5.3-codex issues no environment queries at all and decides directly from the source-code context. Function introspection is rare across the board, consistent with our observation that function behavior is typically inferrable from the surrounding call site without a separate lookup.

Every strategy in the menu is used by at least one model. For merging, `all` (allocation-site collapse) is the dominant choice for most models, with GPT-4o-mini as the notable exception — it picks `field_sensitive` 57.7% of the time. Qwen3-32B has the most balanced merging mix and exercises every strategy in the menu. `role`-based merging is rare across the GPT-5 family ($\leq 1\%$) but reaches 5.4% for GPT-4.1-mini and 6.9% for Qwen3-32B. For widening, `depth` is the most common choice across all models, with `field_value` the typical second choice; GPT-4o-mini is again an outlier, picking `depth` 99.4% of the time. The 10-query tool-call budget is rarely exhausted: 7.9% of decisions for GPT-4o-mini and 0% for every other model.

*Table 7.* Per-model environment query distribution. Counts are totals across 10 runs on the 53-benchmark suite; percentages are taken over total environment queries (excluding the three required strategy-selection calls).

| Model | info_var | execute_loop | info_function | Total | Mix (%) |
|---|---|---|---|---|---|
| GPT-4o-mini | 1,246 | 641 | 54 | 1,941 | 64.2 / 33.0 / 2.8 |
| GPT-4.1-mini | 255 | 333 | 0 | 588 | 43.4 / 56.6 / 0.0 |
| Qwen3-32B | 2 | 2,710 | 0 | 2,712 | 0.1 / 99.9 / 0.0 |
| o4-mini | 21 | 5 | 3 | 29 | 72.4 / 17.2 / 10.3 |
| GPT-5.3-codex | 0 | 0 | 0 | 0 | — |
| GPT-5.4 | 2,084 | 23 | 16 | 2,123 | 98.2 / 1.1 / 0.8 |
| GPT-5.4-mini | 18 | 0 | 1 | 19 | 94.7 / 0.0 / 5.3 |

*Table 8.* Per-model merging strategy distribution (% of merge decisions across 10 runs on the 53-benchmark suite).

| Model | all | recency | role | field_sensitive |
|---|---|---|---|---|
| GPT-4o-mini | 6.7 | 33.3 | 2.3 | 57.7 |
| GPT-4.1-mini | 51.6 | 43.0 | 5.4 | 0.0 |
| Qwen3-32B | 65.5 | 10.8 | 6.9 | 16.7 |
| o4-mini | 61.2 | 37.7 | 1.1 | 0.0 |
| GPT-5.3-codex | 53.9 | 46.0 | 0.1 | 0.0 |
| GPT-5.4 | 64.0 | 36.0 | 0.1 | 0.0 |
| GPT-5.4-mini | 96.2 | 3.0 | 0.9 | 0.0 |

*Table 9.* Per-model widening strategy distribution (% of widen decisions across 10 runs on the 53-benchmark suite).

| Model | depth | field_value | none | all |
|---|---|---|---|---|
| GPT-4o-mini | 99.4 | 0.2 | 0.0 | 0.4 |
| GPT-4.1-mini | 51.3 | 32.2 | 8.5 | 8.0 |
| Qwen3-32B | 40.1 | 6.7 | 33.0 | 20.2 |
| o4-mini | 51.6 | 27.1 | 13.8 | 7.5 |
| GPT-5.3-codex | 62.7 | 33.8 | 2.4 | 1.1 |
| GPT-5.4 | 59.2 | 37.0 | 3.1 | 0.7 |
| GPT-5.4-mini | 50.3 | 31.9 | 15.7 | 2.1 |

# F. LLM Usage

We used a large language model (ChatGPT, GPT-5, OpenAI) to assist with polishing the writing and improving clarity of exposition. The model was not used to design the methodology, conduct experiments, or generate results. All technical contributions, data analysis, and conclusions are the authors' own.

