# OpenReview forum: "ABSINT-AI: Agentic Heap Abstractions for Abstract Interpretation"
_ICML.cc/2026/Conference — ICML 2026 regular_

### Official Review · Reviewer_2pHM · 2026-03-12

**Soundness:** 4
**Presentation:** 4
**Significance:** 4
**Originality:** 3
**Overall Recommendation:** 6
**Confidence:** 5

**Summary:**

This paper presents ABSINT-AI, a novel framework that integrates large language models (LLMs) into abstract interpretation for self contained JavaScript static analysis. The core contribution is an agentic LM-guided approach of heap usage analysis. Unlike traditional static analyzers that rely on uniform, hard-coded heap abstractions, ABSINT-AI employs an LM agent to decide when to expand and when to merge. Its architectural design preserves the formal soundness guarantees of abstract interpretation while allowing semantic cues (e.g., variable naming conventions, access patterns) to guide abstraction decisions. The authors evaluate ABSINT-AI on a soundness-critical task—detecting null/undefined property dereferences and absent property reads—demonstrating up to 34% reduction in false positives compared to fixed-abstraction baselines (TAJS and WALA) while maintaining soundness. Ablations show that the interactive, agentic nature of the approach is crucial: non-agentic LM predictions increase false positives by 25%, and symbolic abstraction without LM guidance increases them by 88%.

**Compliance With Llm Reviewing Policy:**

Affirmed.

**Final Justification:**

The rebuttal addresses my key concern. I raised the score.

**Key Questions For Authors:**

The benchmark intentionally excludes programs with substantial external library dependencies. Is it possible to do more experiments with at least some simple library calls?

**Limitations:**

The authors discuss limitations in the paper.

**Strengths And Weaknesses:**

Strengths: In this paper, the core insight, that LLMs can provide semantic guidance for abstraction decisions while preserving soundness through architectural constraints, is well-executed. The technical design is principled: by restricting LM's behavior to a set of predefined actions and preventing direct state manipulation, the authors achieve a clean separation between learning-based decision-making and reliable formal reasoning. In the experiment, this neurosymbolic approach outperform previous pure symbolic methods. In the evaluation, the authors compare against two established baselines (TAJS and WALA), conduct meaningful ablations isolating the contribution of agentic interaction, and demonstrate that intersection-based aggregation across multiple runs can further improve precision without compromising soundness. The case study on Conway's Game of Life effectively illustrates how semantic understanding enables precise abstraction in real programs. The writing is generally clear, and the methodology is described with sufficient detail to understand the approach.

Weaknesses: The most significant limitation is the small scale of the evaluation. The benchmark consists of only 17 JavaScript programs, manually curated to be "self-contained" and avoid excessive use of built-in libraries. This severe restriction raises serious concerns about generalizability: modern JavaScript development heavily relies on external libraries (npm packages, DOM APIs, framework internals), and the intentional exclusion of such programs means the evaluation may not reflect real-world applicability. Also, the case study Conways's Game of Life is a very simple program with only integers involved. The authors note that prior work like TAJS evaluated on only 8 programs, but this does not justify the limited scope—rather, it highlights a broader methodological weakness in the field that this work does not overcome.

---

> ### Author Rebuttal · Authors · 2026-03-31
>
> Thank you for the thorough review and for recognizing the principled design of our neurosymbolic approach. We address the evaluation scope concern below.
>
> ### Weakness 1: The most significant limitation is the small scale of the evaluation. The benchmark consists of only 17 JavaScript programs, manually curated to be "self-contained" and avoid excessive use of built-in libraries.
>
> We have expanded our benchmark suite from 17 to 53 programs. The new benchmarks include 23 real-world open-source npm libraries such as PapaParse (CSV parser, 1,650 LOC), Showdown (Markdown converter, 1,680 LOC), Validator.js (string validation, 1,885 LOC), and JSBigNum (1,357 LOC). In total, 9 benchmarks exceed 1,000 LOC and 6 exceed 1,500 LOC — well beyond the prior maximum of 577 lines. All results and artifacts will be released.
>
> **Reproduced results on expanded benchmarks (Table 1):**
>
> | Method | Mean FP | Intersection FP |
> |--------|---------|-----------------|
> | WALA | 692 | — |
> | TAJS | 711 | — |
> | ABSINT-AI (GPT-4.1-mini, 10 runs) | 618.7 | 555 |
> | No-agent (direct prediction) | 645.7 | 572 |
>
> TAJS crashes on 11/53 benchmarks; for those we conservatively add ABSINT-AI's own FP count (525 + 185.8 = 711).
>
>
> ### Question 1: The benchmark intentionally excludes programs with substantial external library dependencies. Is it possible to do more experiments with at least some simple library calls?
>
> Yes — we expanded our benchmarks to include real-world libraries with substantial built-in API usage, and will include these in the revision.
>
> **Experiment: Expanded Benchmarks with Library Usage.** We expanded from 17 to 53 benchmarks, including 23 real-world open-source npm libraries such as PapaParse (CSV parser, 1,650 LOC), Showdown (Markdown converter, 1,680 LOC), Validator.js (string validation, 1,885 LOC), Mustache.js (template engine, 618 LOC), JSBigNum (arbitrary-precision arithmetic, 1,357 LOC), and UAParser (user-agent parsing, 908 LOC). Of the 53 benchmarks, 15 use DOM APIs as core logic, and 22 use RegExp or Date built-ins. All results in W1 above (Table 1) are reported on this expanded set. ABSINT-AI successfully analyzes all 53.

---

> > ### Author Rebuttal · Reviewer_2pHM · 2026-04-04
> >
> > Thank you for the rebuttal. My concerns are all addressed. I will raise my score.

---

### Official Review · Reviewer_6M2J · 2026-03-12

**Soundness:** 3
**Presentation:** 3
**Significance:** 3
**Originality:** 3
**Overall Recommendation:** 5
**Confidence:** 4

**Summary:**

The paper introduces a JavaScript Static Analyzer that makes widening and merging decisions using LLMs. The benefit of this approach is that the model can make decisions per object based on context from the surrounding code and future or past usage. The authors demonstrate that this effectively reduces false positives on a dataset of 17 JavaScript code bases compared to two prior baselines.

**Compliance With Llm Reviewing Policy:**

Affirmed.

**Final Justification:**

The authors cleanly ablated their proposed method and considerably increased the size of their evaluation benchmark, even beyond the standard in the field (in response to the request of another reviewer). Application of LLMs for static analysis in the way presented is IMO established as quite promising through this work (at least complementary to existing methods). I disagree with other reviewers that remaining False Positives present a challenge, because abstract interpretation results can meaningfully complement each other as long as they remain sound.

**Key Questions For Authors:**

- Why were the method choices done as presented and what happens if they are ablated?
- Do the most recent LLMs (smaller and more capable) improve the proposed method even further? Does the method thus generally benefit from the overall scaling and capability trend of LLMs?
- Does the method apply to other languages?

**Limitations:**

yes

**Strengths And Weaknesses:**

Strengths

- *Originality*: I like that the authors effectively introduce reliable soundness guarantees by limiting the scope of actions that the LLM can perform. This leverages the strong reasoning and understanding capabilities of the LLM while maintaining guarantees. It appears that specifically for LLMs, no prior work explored using them to guide abstract interpretation decisions.
- *Significance*: The field of abstract interpretation is not very relevant currently, due to the high number of false positives in interpreters. This work makes an interesting advance to soundly improve this limitation of interpreters and might inspire similar adaptations in other uses.

Weaknesses

- *Significance*: The authors inspect only JavaScript. This is unfortunate because LLMs are quite general and should be applicable to a wide range of languages to improve their abstract interpretation. I think its fine for an exploratory work however, since there appears to be some non-neglibible effort involved in designing the guidable abstract interpretation.
- *Significance*: A significant downside of this work is the associated cost. Even if the harness inefficiency was ignored, the LLM inference part of the method requires 9 times as long as the baselines. We can expect LLMs to become more capable and faster in the future however, which might improve this downside.
- *Soundness*: The choices in the methods section appear somewhat arbitrary: 10 read only queries, fixed widening and merging operators. Do these cover all possibilities, are they representative, sometimes used? What is the impact of changing these?
- *Originality*: I think the authors miss much related work in using machine learning to improve abstract interpreter precision, for example [1]
- *Soundness*: While the selected two works appear to be commonly used libraries for JavaScript abstract interpretation, the authors miss other related works on static analysis, such as [2,3]
- *Presentation*: In line 418 the authors present a case study to highlight the improvement of their LLM abstraction over baselines. However the wording is general - do the baselines TAJS and WALA correct abstract this structure or do they over abstract?
- *Soundness*: In 4.3 the authors abort the analysis if it does not converge. While this is reasonable, the output is hard to compare with the other baselines because an aborted run is not sound. Since it anyways overreports FPs this is a unique case where this works, but it reads a bit confusing.

Typo in line 281 TAJS is a performs

[1] Singh et al, Fast Numerical Program Analysis with Reinforcement Learning, https://files.sri.inf.ethz.ch/cav18-paper259.pdf
[2] Wang et al, Efficient Abstract Interpretation via Selective Widening, OOPSLA 2025
[3] Dewey et al, A Parallel Abstract Interpreter for JavaScript, IEEE CGO 2015

---

> ### Author Rebuttal · Authors · 2026-03-31
>
> Thank you for your detailed and constructive feedback! We address each concern below.
>
> ### W1/Q3: Only JavaScript evaluated.
>
> We chose JavaScript because it has arguably the most dynamic heap model of any mainstream language, making heap abstraction particularly challenging. The agent interface (Algorithm 1) is language-agnostic (parameterized by abstraction strategies and tool queries, not the target language) so any abstract interpreter with configurable heap summarization could adopt this approach. The agentic layer is small; the bulk of the effort is the underlying interpreter (~12,000 lines for ES5).
>
> ### W2: Runtime.
>
> We acknowledge the cost limitation. As the first work applying LLMs to guide abstract interpretation, we focused on establishing that LLMs can soundly improve precision (up to 34% FP reduction). Optimizing runtime is an important next step that will benefit from both continued drops in LLM inference costs and community effort from the PL/SE side as researchers develop program analysis tooling around LLM agents. Promising avenues:
>
> 1. **Fine-tuned local models.** Distill the agent's decisions into a smaller model. Our strategy menu is a small discrete set, making this a standard classification task with negligible latency.
> 2. **Selective invocation.** Only invoke the LLM when default symbolic abstractions fail to converge, eliminating the majority of LLM calls.
> 3. **Parallelism.** Runs are fully independent (trivially parallelizable), and within a run, independent loops could be analyzed concurrently.
>
>
> ### W3/Q1: Arbitrary method choices — impact of ablating strategies and query budget?
>
> Our operators draw from established techniques (recency abstraction, allocation-site merging) and LLM-specific strategies like role-based merging, which partitions objects by semantic role and is difficult to implement symbolically. They do not cover all abstractions (e.g., relational); incorporating more complex strategies is future work we will discuss in the revision.
>
> **Experiment: Query Budget Sensitivity.** Budget varied from 0 to 15 (10 runs each, 53 benchmarks):
>
> | Budget | Mean FP |
> |--------|---------|
> | 0 | 645.7 |
> | 1 | 628.5 |
> | 3 | 633.0 |
> | 5 | 631.7 |
> | 7 | 621.4 |
> | 10 (default) | 613.6 |
> | 15 | 614.2 |
>
> Performance improves steadily as budget increases, from 645.7 (no queries) to 613.6 at the default of 10, confirming that environment queries meaningfully improve precision beyond source code context alone.
>
> **Experiment: Strategy Ablation.** We removed each merging/widening strategy individually and measured the impact:
>
> | Removed Strategy | Mean FP | Delta vs Full |
> |-----------------|---------|---------------|
> | (none — full system) | 618.7 | — |
> | No recency (merge) | 637.6 | +18.9 |
> | No role (merge) | 630.0 | +11.3 |
> | No field_value (widen) | 642.5 | +23.8 |
> | No depth (widen) | 627.8 | +9.1 |
>
> Every strategy contributes. field_value widening (+23.8 FP) and recency merging (+18.9 FP) have the largest impact.
>
> ### W4: Missing related work [1,2,3].
>
> Thank you. Singh et al. (2018) use RL to select abstract transformers in numerical domains (Polyhedra) for speed (up to 515x) while maintaining precision — similar in using learned policies to guide analysis, but with hand-crafted features on numerical domains rather than LLM-guided heap abstraction. Wang et al. (OOPSLA 2025) selectively widen only variables in value-flow cycles, reducing analysis time by 41%. Dewey et al. (CGO 2015) parallelize JavaScript abstract interpretation, orthogonal to our work. We will include all three in the revision.
>
> ### W5: Case study — do TAJS/WALA over-abstract?
> We ran both TAJS and WALA on Game of Life. Both over-abstract `cell_state`, prematurely merging array rows and losing row-level distinctions, resulting in additional false positives.
>
> ### W6: Aborted runs are not sound — confusing to compare with baselines.
>
> We will clarify the presentation in the revision.
>
>
> ### Q2: Does the method benefit from newer/more capable LLMs?
>
> We evaluated five recent models to investigate this. We will include these results in the revision.
>
> **Model Comparison** (53 benchmarks, 10 runs each):
>
> | Model | Mean FP | loop step |
> |-------|---------|-----------|
> | gpt-4.1-mini | **618.7** | 333 |
> | o4-mini | 631.8 | 5 |
> | gpt-5.3-codex | 638.0 | 0 |
> | gpt-5.4 | 638.1 | 23 |
> | gpt-5.4-mini | 640.3 | 0 |
>
> Surprisingly, gpt-4.1-mini remains the best-performing model. A root cause analysis reveals this is because it makes significantly more use of the abstract loop step tool (333 calls vs near-zero for newer models).
>
> We attribute this to prompt sensitivity: our prompt discourages repeated loop execution, and newer models over-comply. We will tune prompts for newer models in the revision. As models improve, incorporating more complex abstractions where stronger reasoning has a larger impact is a natural next step.

---

> > ### Author Rebuttal · Reviewer_6M2J · 2026-04-03
> >
> > I thank the reviewers for their extensive answers. I am adjusting my score accordingly.

---

### Official Review · Reviewer_WYPE · 2026-03-12

**Soundness:** 1
**Presentation:** 1
**Significance:** 1
**Originality:** 2
**Overall Recommendation:** 2
**Confidence:** 4

**Summary:**

The paper presents an LLM-aided abstract interpreter ABSINT-AI, which can generate adaptive heap abstraction strategies for related static analysis tasks. Experiments on null dereference detection manifest the effectiveness and efficiency of this approach to certain extent.

**Compliance With Llm Reviewing Policy:**

Affirmed.

**Final Justification:**

The paper requires substantial revision to address its issues in presentation and scalability, which are not acceptable at this stage. Furthermore, its methodological weakness in applying LM agents for sound static analysis remains unresolved. Hence, my score remains the same.

**Key Questions For Authors:**

1. How many benchmarks can each approach handle respectively?
2. What approach does Mean stand for in Table 1?
3. What's the purpose of 4.4, which does not reflect the above motivating observation？

**Limitations:**

Typical threats to the validity of this work needs to be discussed. The scalability limitation discussed is rather a weakness that needs to be taken care of in the first place.

**Strengths And Weaknesses:**

Strength
+ Interesting observation on how LLMs can fit static analysis by leverage their natural language processing capabilities in program context.

Weaknesses
- The paper has been poorly written with redundant wording, while leaves the methodological details absent or scattered throughout the paper. For example, it is not clear how the main workflow actually proceeds step by step.
- It is claimed as a soundness guarantee in this paper to restrain LLMs from performing downstream tasks directly. However, this only means no false positive may be introduced by LLMs, but does not guarantee no false positive at all. The experimental results also confirm that false positives still remain.
- Intersecting results across multiple runs actually leads to less positives reported, hence less false positives. However, this indeed entails a risk of missing bugs (i.e., false negatives).
- The experiments reported are rather small-scale, with limited or uncertain generalizability for other benchmarks or static analysis tasks. It is not clear how a static analyzer can be plugged into the implemented ABSINT-AI framework.

---

> ### Author Rebuttal · Authors · 2026-03-31
>
> Thank you for your feedback and for recognizing the interesting observation on how LLMs can fit static analysis! We address each concern below.
>
> ### W1: Unclear methodology and workflow presentation.
>
> We acknowledge that not all readers will have a program analysis background and will revise accordingly. ABSINT-AI is a static analyzer that uses an LLM agent to guide how it *summarizes heap objects*. The workflow:
>
> 1. **Abstract interpretation begins.** The interpreter walks the program symbolically, tracking what values variables and object fields may hold — a standard abstract interpretation pipeline.
> 2. **Agent invoked at loop heads.** When the interpreter reaches an unbounded loop (where summarization decisions critically affect precision and termination), it pauses and invokes the LLM agent.
> 3. **Agent inspects and decides.** The agent receives the source code context (variable names, access patterns, surrounding logic) and may optionally query the analysis state (e.g., inspect a variable's current abstract value or step through a loop iteration). It then selects, for each heap object, a summarization strategy from a predefined menu of sound options.
> 4. **Interpreter resumes.** The interpreter applies the agent's chosen strategies and continues to a fixpoint. If convergence fails, the agent is re-queried for alternative strategies.
> 5. **Bug reporting.** Any property access on a potentially null/undefined value is reported as a warning.
>
> Crucially, the agent only selects strategies and never manipulates program state directly. All semantic computation remains in the interpreter, preserving soundness by construction. We will restructure the presentation to foreground this pipeline more clearly.
>
>
> ### W2: Soundness claim only prevents false positives from LLMs, not false positives overall.
> We respectfully clarify: soundness guarantees the absence of false *negatives*, not false positives. We state this explicitly (line 47-48 in the PDF) and formally in Appendix A (line 616). The LLM's choices can only affect precision (number of false positives), and can never cause the analysis to miss a real bug. We recognize that "soundness" can mean different things in different communities; we will add a precise definition in the introduction to avoid confusion.
>
> ### W3: Intersection may miss bugs (false negatives).
> We addressed this directly in Section 3.3 (lines 312-319). Each run is individually sound, so every real bug appears in every run. The intersection can therefore only remove false positives.
>
> ### W4: Small-scale evaluation with limited generalizability.
>
> We expanded from 17 to 53 programs, including 23 real-world npm libraries (PapaParse 1,650 LOC, Showdown 1,680 LOC, Validator.js 1,885 LOC, JSBigNum 1,357 LOC). 9 benchmarks exceed 1,000 LOC, 6 exceed 1,500 LOC. Updated results:
>
> | Method | Mean FP | Intersection FP |
> |--------|---------|-----------------|
> | WALA | 692 | — |
> | TAJS | 711 | — |
> | ABSINT-AI (GPT-4.1-mini, 10 runs) | 618.7 | 555 |
> | No-agent (direct prediction) | 645.7 | 572 |
>
> TAJS crashes on 11/53; for those we conservatively add ABSINT-AI's own FP count (525 + 185.8 = 711).
>
> ### W5: Unclear how other analyzers can adopt this framework.
> The agent interface (Algorithm 1) is language-agnostic, and is parameterized by available abstraction strategies and tool queries, not the target language. Any abstract interpreter with configurable heap summarization could adopt this approach. We chose JavaScript because it has among the most dynamic heap models of any mainstream language. The agentic component itself is a small, modular layer; the bulk of the effort is the underlying interpreter (~12,000 lines of Python), which is inherent to any sound static analysis project.
>
> ### Q1: How many benchmarks can each approach handle?
> On the expanded 53-benchmark suite: WALA and ABSINT-AI successfully analyze all 53 benchmarks. TAJS crashes on 11/53 (including several real-world libraries such as PapaParse, Validator.js, Mustache.js, and JSBigNum). See the W4 above for the full comparison table.
>
> ### Q2: What does "Mean" refer to in Table 1?
> As stated in Section 4.2 (line 368-369), "Mean" refers to the arithmetic mean of false positives across 10 runs of ABSINT-AI for each model. This captures the expected performance of a single run.
>
> ### Q3: What's the purpose of Section 4.4?
> Section 4.4 directly illustrates the core motivation: different heap objects require different abstraction strategies. The agent distinguishes the structure of the cell_state array (preserved) from its integer values (abstracted) — the same semantically-informed, per-object abstraction shown in the Section 2 motivating example.

---

> > ### Author Rebuttal · Reviewer_WYPE · 2026-04-03
> >
> > My score remains as is, since a) the presentation quality is weak: apart from poor writing, *two* tiny motivating examples, and even in the software engineering community it is necessary to define clearly and explicitly what are the positive cases; b) the technical strength is weak, even from the software engineering perspective: the major soundness claim directly comes from leveraging traditional sound static analysis techniques, while the LM agent itself does not substantively contribute to or struggle with it; in the meanwhile, the gain of using the LM agent is limited considering the overhead of calling it repeatedly, esp., as shown in Figure 11, the LM agent may noticeably increase false positives.
> >
> > Besides, adding (53-17)/17=2.1x more experiments during the rebuttal period appears very unusual.

---

> > > ### Author Response · Authors · 2026-04-06
> > >
> > > Thank you for your feedback! Our responses are below.
> > >
> > > ### (a) Presentation: writing and motivating examples.
> > >
> > > We acknowledge that writing quality can be improved and will revise accordingly. The two examples (motivating example in Figure 1 and case study in Section 4.4) were kept small due to the 8-page limit. We will add a larger, more detailed example to the appendix in the revision.
> > >
> > > ### (b) Technical strength: soundness comes from traditional techniques, LLM does not substantively contribute.
> > >
> > > Preserving soundness by construction is precisely the contribution — we designed the architecture so that the LLM operates through a bounded, read-only interface where its errors can only affect precision (more false positives), never correctness. This is what makes it safe to embed an LLM inside a sound analysis, which no prior work has done.
> > >
> > > The LLM substantively contributes to precision. The no-agent ablation (Section 4.3, Figure 5) shows that removing the agent increases false positives by 25%. The symbolic baseline (which uses the same abstraction strategies but without agentic selection) produces 28.6% more false positives than TAJS. The agent's ability to reason about program context and select per-object strategies is what closes this gap. In particular, role-based merging (22.3% of agent decisions) requires semantic understanding of field names and value meanings to partition objects by role. This is inherently a natural language reasoning task that traditional symbolic techniques struggle to perform.
> > >
> > > ### (c) Figure 11: LLM agent may increase false positives on some benchmarks.
> > >
> > > We agree that there are individual benchmarks where TAJS outperforms ABSINT-AI. We are not claiming that using an LLM is categorically better on every benchmark. TAJS's hand-engineered heuristics are well-tuned and work well in many cases. Our claim is that using an LLM *overall* improves precision, as shown in Table 1: across the full benchmark suite, ABSINT-AI achieves fewer total false positives than both TAJS and WALA.

---

### Official Review · Reviewer_XsxP · 2026-03-12

**Soundness:** 2
**Presentation:** 3
**Significance:** 2
**Originality:** 3
**Overall Recommendation:** 4
**Confidence:** 4

**Summary:**

This paper introduces ABSINT-AI, a framework that integrates an LLM agent within an abstract interpreter to guide heap abstraction decisions for JavaScript programs. The key idea is that the agent operates through a constrained, read-only interface, selecting among a fixed menu of sound abstraction strategies, so that soundness of the underlying abstract interpretation is preserved by construction. The downstream task is detecting null/undefined dereferences and absent property accesses

**Compliance With Llm Reviewing Policy:**

Affirmed.

**Final Justification:**

Authors have addressed my concerns.

**Key Questions For Authors:**

1. All 17 benchmarks were manually inspected to confirm that the property of interest (absence of unsafe property accesses) holds → how does the tool perform if the benchmark actually has a bug? With the intersection approach all the runs have to detect the bug for it to show up, any run where the LLM selects an abstraction that masks the bug would cause it to be silently dropped from the final result?
2. Section 3.2 states that if a fixpoint is not reached within five iterations, the agent is re-queried for new abstraction strategies. It is unclear whether the agent has access to its prior decisions and the strategies it already attempted. Without this context, the agent may simply repeat the same failing choices.
3. ABSINT-AI takes 500 seconds → is this the time for the intersection runs which achieves the best false positive rate or the single run number? If this is not the intersection run time, then how much time does it take for the full intersection?

**Limitations:**

Yes

**Strengths And Weaknesses:**

Strengths:
1. I really liked the idea of constraining the LLM to select among a fixed, pre-verified set of abstraction operators while leaving all semantic computation and state transitions to the abstract interpreter. This separation of concerns means LLM errors can only affect precision (more false positives), but cannot compromise soundness

Weakness:
1. The evaluation uses only 17 programs with a maximum of 577 lines of code. This is particularly concerning given the limitation section states that the tool does not scale beyond 2,000+ lines, yet no benchmark approaches this limit
2. The approach relies heavily on prompting LLMs. However, no detailed prompts are shown in the main text or appendix for reproducibility. Particularly for the single-shot ablation experiment where the exact prompt formulation directly affects the performance.
3. Beyond metrics the evaluation provides no analysis of which abstraction strategies the agent actually selects across benchmarks, nor how the tool query budget is utilized in practice. Concretely: how many queries are issued on average per decision, which query types dominate (variable inspection, function introspection, or abstract loop step), and how often is the 10-query budget exhausted? Without this, it is hard to assess whether the agentic interaction is being used.

---

> ### Author Rebuttal · Authors · 2026-03-31
>
> Thank you for your thoughtful evaluation! We address each concern below.
>
>
> ### W1: Small evaluation.
>
> We expanded from 17 to 53 programs, including 23 real-world npm libraries (PapaParse 1,650 LOC, Showdown 1,680 LOC, Validator.js 1,885 LOC, JSBigNum 1,357 LOC). 9 benchmarks exceed 1,000 LOC, 6 exceed 1,500 LOC. Updated results:
>
> | Method | Mean FP | Intersection FP |
> |--------|---------|-----------------|
> | WALA | 692 | — |
> | TAJS | 711 | — |
> | ABSINT-AI (GPT-4.1-mini, 10 runs) | 618.7 | 555 |
> | No-agent (direct prediction) | 645.7 | 572 |
>
> TAJS crashes on 11/53; for those we conservatively add ABSINT-AI's own FP count (525 + 185.8 = 711).
>
>
> ### W2: No prompts shown for reproducibility.
>
> We will include all prompts and tool schemas in the appendix. The key prompts are:
>
> 1. **System prompt (agentic mode):** Instructs the agent to decide which heap allocation sites need summarization for loop convergence. The agent gathers information via tool queries before selecting sites, constrained to respond only via tool calls. A budget of 10 tool queries prevents excessive querying.
> 2. **System prompt (no-agent ablation):** Same task description, but without access to environment query tools — the agent must select allocation sites based solely on the source code context.
> 3. **Merging strategy prompt:** For each selected allocation site, the agent chooses from the merging strategies in section 3.2. The agent receives the surrounding code, loop body, and current abstract value.
> 4. **Widening strategy prompt:** For each selected allocation site, the agent chooses from the widening strategies in section 3.2. Same context as above.
>
>
> ### W3: No analysis of agent query behavior or strategy selection.
>
> This is a great suggestion. We performed a detailed analysis of agent behavior and will include it in the paper/appendix.
>
> **Experiment: Behavior Statistics.** Across 10 runs on 53 benchmarks (1,929 decisions, 7,155 tool calls):
>
> - **Environment queries (458 total):** variable inspection 67.7% (310), abstract loop step 28.8% (132), function introspection 3.5% (16).
> - **Merging strategies:** all (56.3%), role (22.3%), recency (19.9%), field_sensitive (1.5%).
> - **Widening strategies:** depth (52.8%), field_value (32.5%), none (7.6%), all (7.1%).
> - **LLM time fraction:** 47.7% of total runtime.
>
> All strategies are actively used. Function introspection is rare (3.5%) because function behavior is typically inferrable from its name and calling context. field_sensitive merging is seldom needed (1.5%) because objects at the same allocation site tend to share the same field set. The default budget of 10 was never exhausted in our experiments. Removing the agent (no-agent ablation) increases mean FP from 618.7 to 645.7.
>
> **Experiment: Query Budget Sensitivity.** Budget varied from 0 to 15 (10 runs each, 53 benchmarks):
>
> | Budget | Mean FP |
> |--------|---------|
> | 0 | 645.7 |
> | 1 | 628.5 |
> | 3 | 633.0 |
> | 5 | 631.7 |
> | 7 | 621.4 |
> | 10 (default) | 613.6 |
> | 15 | 614.2 |
>
> Performance improves steadily as budget increases, from 645.7 (no queries) to 613.6 at the default of 10, confirming that environment queries meaningfully improve precision beyond source code context alone.
>
>
> ### Q1: How does the tool perform on programs with real bugs? Can intersection drop true bugs?
>
> Soundness guarantees that every real bug is reported in every run — the analysis over-approximates all reachable states, so abstraction choices can only affect which false positives appear, never whether true bugs are detected. The intersection across runs can therefore only remove false positives, never true bugs.
>
> **Experiment: Bug Injection.** We injected real bugs into 4 benchmarks (null dereferences and absent property accesses) and ran ABSINT-AI 10 times each. All injected bugs were detected in 100% of runs (42/42 total) and in every intersection.
>
>
> ### Q2: Does the agent have memory of prior failed strategies on re-query?
>
> Thank you for raising this. In the submitted version, the agent does not receive context about prior failed strategies on re-query.
>
> **Experiment: Agent Memory.** We implemented a variant that passes the agent its prior strategy selections and convergence failure context when re-queried. Results over 10 runs on 53 benchmarks:
>
> | Variant | Mean FP | Intersection FP |
> |---------|---------|-----------------|
> | Without memory | 618.7 | 555 |
> | With memory | 614.6 | 557 |
>
> The improvement is marginal, suggesting the agent already selects diverse strategies due to LLM sampling temperature. We will include this in the revision.
>
>
> ### Q3: Is 500s the single-run or intersection time?
> The 500 seconds reported is for a single run. The full 10-run intersection takes ~5,000s sequentially. However, each run can be trivially parallelized, reducing wall-clock time back to ~500 seconds.

---

> > ### Author Rebuttal · Reviewer_XsxP · 2026-04-04
> >
> > Thank you for the detailed response! Appreciate all the efforts the authors put into getting all the numbers in such a short time frame.
> >
> > W1 -- The earlier evaluation showed a big gap between WALA and TAJS. But the new numbers are some what close. Could you clarify what could be the reason for this?
> >
> > Q3: is it 10 runs or 30 runs? (line 346 --  intersecting all 30 runs across all models yields a 13% reduction in false positive)

---

> > > ### Author Response · Authors · 2026-04-05
> > >
> > > Thank you for the kind words and the follow-up questions!
> > >
> > > ### W1 follow-up: Why are WALA and TAJS numbers close?
> > >
> > > Upon investigation, we found that our WALA harness was configured to check only absent property accesses, not null/undefined dereferences — so it was evaluating a narrower scope of warnings than TAJS and ABSINT-AI. After updating the configuration to include all warning types, WALA's FP count increases from 692 to 1327:
> > >
> > > | Method | Mean FP | Intersection FP |
> > > |--------|---------|-----------------|
> > > | WALA | 1327 | — |
> > > | TAJS | 711 | — |
> > > | ABSINT-AI (GPT-4.1-mini, 10 runs) | 618.7 | 555 |
> > > | No-agent (direct prediction) | 645.7 | 572 |
> > >
> > > This gap between WALA and TAJS is consistent with the expected difference between a flow-insensitive analysis (WALA) and a flow-sensitive one (TAJS). Thank you for prompting us to investigate this discrepancy!
> > >
> > > ### Q3 follow-up: 10 runs or 30 runs?
> > >
> > > Apologies for the confusion. In the rebuttal, we report 10 runs with a single model (GPT-4.1-mini). The "30 runs" in line 346 of the paper refers to 10 runs × 3 models (GPT-4, Qwen, Llama), and the intersection there is taken across all 30. The timing experiments were done with GPT-4.1-mini, where all runs across the benchmark can be parallelized.
> > >
> > > The locally-hosted models (Qwen, Llama) each take \~1.5× longer per run (\~750s vs \~500s) and cannot be parallelized across the benchmark suite on our single server, unlike API-based models. With API models, all 10 runs can be launched in parallel so the full intersection still takes ~500s wall-clock. With the locally-hosted models, the 10 runs must be sequential due to our local compute constraints, totaling ~7,500s per model.

---

### Decision · Program_Chairs · 2026-04-30

**Decision:**

Accept (regular)

**Comment:**

Reviewers agreed that this paper tackles an interesting problem in LLM-supported static program analysis. The paper contributes a novel interesting methodology using LLMs to guide semantics abstraction decisions in static program analysis based on abstract interpretation. The paper shows the strength of this approach on several benchmarks. Most of the reviewers and the area chair agree that the results of this paper are solid contribution for the ICML community, and recommend acceptance.